



# Quantification of Isomer-Resolved Iodide CIMS Sensitivity and Uncertainty Using a Voltage Scanning Approach

Chenyang Bi[1], Jordan E. Krechmer[2], Graham O. Frazier[1], Wen Xu[2], Andrew T. Lambe[2], Megan S. Claflin[2], Brian M. Lerner[2], John T. Jayne[2], Douglas R. Worsnop[2], Manjula R. Canagaratna[2], Gabriel Isaacman-VanWertz[1]

[1]Department of Civil and Environmental Engineering, Virginia Tech, Blacksburg, Virginia, 24060, USA
[2]Aerodyne Research Inc, Billerica, Massachusetts, 01821, USA

*Correspondence to*: Gabriel Isaacman-VanWertz (ivw@vt.edu)

**Abstract.** Chemical ionization mass spectrometry (CIMS) using iodide as a reagent ion has been widely used to classify organic compounds in the atmosphere by their elemental formula. Unfortunately, calibration of these instruments is challenging due to a lack of commercially available standards for many compounds, which has led to the development of methods for estimating CIMS sensitivity. By coupling a Thermal desorption Aerosol Gas chromatograph (TAG) simultaneously to a flame ionization detector (FID) and an iodide CIMS, we use the individual particle-phase analytes, quantified by the FID, to examine the sensitivity of the CIMS and its variability between isomers of the same elemental formula. Iodide CIMS sensitivities of isomers within a formula are found to generally vary by one order of magnitude with a maximum deviation of two orders of magnitude. Furthermore, we compare directly measured sensitivity to a method of estimating sensitivity based on declustering voltage (i.e., "voltage scanning"). This approach is found to carry high uncertainties for individual analytes (half to one order of magnitude), but represents a central tendency that can be used to estimate the sum of analytes with reasonable error (~30% differences between predicted and measured moles). Finally, GC retention time, which is associated with vapor pressure and chemical functionality of an analyte, is found to qualitatively correlate with iodide CIMS sensitivity, but the relationship is not close enough to be quantitatively useful and could be explored further in the future as a potential calibration approach.

## 1 Introduction

Air pollution is ranked as a major risk factor for global illness and death (Stanaway et al., 2018). Exposure to ambient fine particulate matter ($PM_{2.5}$) is associated with severe health outcomes (Burnett et al., 2014; Pope and Dockery, 2006). A substantial fraction of $PM_{2.5}$ is secondary organic aerosol (SOA) that is generated through atmospheric oxidation of volatile organic compounds (VOCs) (Hallquist et al., 2009; Kroll and Seinfeld, 2008; Shrivastava et al., 2017). Characterizing the molecular composition of organics in SOA and precursor gases is crucial for understanding the chemical fate, removal, and ultimately the impact on human and environmental health. However, the complexity of atmospheric mixtures represents a significant analytical challenge (Goldstein and Galbally, 2007; Jimenez et al., 2009; Kroll and Seinfeld, 2008).


High-resolution time-of-flight chemical ionization mass spectrometry (HR-ToF-CIMS) has been widely used to directly sample and characterize gas- and particle-phase organics in ambient and laboratory-generated atmospheres. Chemical ionization offers a relatively "soft" technique in which analytes form ions that do not significantly fragment within the mass

spectrometer. Since the original ions ("parent ions") are preserved for detection by high-resolution mass spectrometer, their elemental formulas can be identified from the accurate mass of detected ions. These instruments consequently classify the diverse atmospheric components by their formulas, though they cannot provide much information regarding molecular structure. A variety of reagent ions are used in atmospheric applications of CIMS, each of which provides selectivity for analytes with a different range of chemical properties, with the most widely used including: iodide for the detection of a wide

range of polar organic compounds (Lee et al., 2014; Slusher et al., 2004), $CF_3O^-$ for oxygenated organics including hydroperoxides (Crounse et al., 2006), acetate for organic acids (Bertram et al., 2011; Brophy and Farmer, 2016), nitrate for highly oxygenated organics (Jokinen et al., 2012; Krechmer et al., 2015), hydronium for VOCs (Yuan et al., 2016), benzene cation for select biogenic VOCs (Kim et al., 2016), and NO+ for branched alkanes, alkyl-substituted aromatics, and other VOCs (Koss et al., 2016). Iodide is frequently used as a reagent ion in CIMS due to its simple ionization chemistry

(Aljawhary et al., 2013; Lee et al., 2014; Pagonis et al., 2019; Riva et al., 2019; Zhang et al., 2018). Iodide forms an adduct with the neutral analyte molecule and the adduct can be used for compound identification and quantification. The iodide-molecule adduct can be easily resolved from any non-adduct ions due to the high negative mass defect of iodine. Therefore, an iodide CIMS enables the online measurement of oxygenated organic compounds with confident classification by elemental formula and high time resolution.


The major limitation of an iodide CIMS is its large range of sensitivities to different molecules, which can range across several orders of magnitude (Iyer et al., 2016) due to variations in binding enthalpies between neutrals and the iodide anion. Quantification of an analyte consequently requires calibration using commercially-available or synthesized chemical standards of the target analytes. However, doing so for many analytes is costly and labor-intensive, and many of the

oxidation products present in ambient atmospheres cannot be efficiently synthesized or isolated as pure compounds (Brophy, 2016). Furthermore, an analyte of interest may exist in the atmosphere alongside other isomers of the same elemental formula, which are not resolved by a mass spectrometer alone, confounding efforts to calibrate using individual analytes. These difficulties have led to the development of empirical approaches to tackle the calibration of atmospheric constituents. In theory, iodide CIMS has a maximum sensitivity dictated by the collision rate of reagent ions (I⁻) with analyte ions,

assuming that any collision forms an adduct. This maximum sensitivity can be calculated based on the interaction time of analyte molecules with reagent ions (Huey et al., 1995; Kercher et al., 2009; Lee et al., 2014). Experimentally, an analyte to which the iodide CIMS is known to be maximally sensitive (typically $N_2O_5$) can be used as a calibrant to determine the observed maximum sensitivity of the instrument, which typically agrees well (Lopez-Hilfiker et al., 2016) or at least within a factor of 4 (Isaacman-Vanwertz et al., 2018) with the theoretical observed maximum sensitivity. However, these methods





estimate only maximum possible sensitivity, while many analytes may not efficiently form iodide adducts, or the formed
      adducts may decompose to generate fewer detectable ions per molecule (Lopez-Hilfiker et al., 2016).

      Quantification based on maximum sensitivity provides only a lower limit on the concentration of observed analytes. To
      refine this quantification method, Iyer et al. (2016) demonstrated through computational work that the binding energy of an
analyte with the reagent ion is log-linearly correlated to observed sensitivity. The binding energy can, in turn, be estimated
      by varying the voltage differentials in the mass spectrometer focusing optics to induce de-clustering (specifically, the voltage
      differential between the skimmer of the first quadrupole and the entrance to the second quadrupole ion guide) (Lopez-
      Hilfiker et al., 2016). Lopez-Hilfiker et al. (2016) showed that "de-clustering scans" or "voltage scans" could empirically
      provide approximate sensitivity of an iodide CIMS, which has since been extended to estimate the sensitivity of other
reagent ion chemistries (i.e., acetate and $NH_4^+$) by modulating various operating conditions to probe product-ion formation
      and stability (Brophy and Farmer, 2016; Zaytsev et al., 2019). However, the quantitative relationships between sensitivity
      and variations in operating conditions are built on a small number of available chemical standards. It is not well known
      whether these relationships hold for the large number of short-lived and complex compounds generated in the atmospheric
      oxidative processes, or how best to validate them for short-lived atmospheric components.
      A further challenge for quantification of CIMS data is that isomers cannot be differentiated because analytes are measured
      only by their elemental formulas. These formulas likely represent multiple molecules, as isomers are found to be prevalent in
      the atmosphere and may vary by orders of magnitude in their CIMS sensitivity (Lee et al., 2014). Based on samples collected
      from a wide range of instruments and environments, Bi et al. (2021b) demonstrated that laboratory-generated samples of
simulated atmospheric oxidation contain many molecules of the same elemental formula – typically 2 to 4, but up to nearly
      20. Previous work has also found very high sample-to-sample and day-to-day variability in molecular-level particle
      composition (Ditto et al., 2018), suggesting a CIMS-detected elemental formula may represent a dynamic and variable set of
      isomers. These isomers, although having the same elemental formula, may have significantly different functional groups and
      detailed chemical structures which consequently determine their physical and chemical properties (Atkinson and Arey, 2003;
Goldstein and Galbally, 2007). Vapor pressure, polarity, reactivity, and compound toxicity are all impacted by the functional
      groups present in a molecule (Arangio et al., 2016), and in some cases by its physical conformation (Atkinson, 2000; Lim
      and Ziemann, 2009). An accurate analysis of the deconvolution of isomers in complex samples is therefore necessary to
      determine the molecular-level composition of the atmosphere, study the formation, transport, and fate of airborne organics,
      and better understand their impacts to global climate and human health. To better apply CIMS instrumentation to these
questions and understand the impacts of changing isomer-composition on calibration, it is important to investigate the
      variability in CIMS sensitivity between isomers.



Isomer-resolved analysis is typically achieved using chromatography techniques. In this work, we focus on gas

chromatography (GC), which has been demonstrated to be an effective way for the online analysis of low-polarity gas-phase

components (Goldan et al., 2004; Goldstein et al., 1995; Helmig et al., 2007; Millet, 2005; Prinn et al., 2000; Vasquez et al.,

2018). More recently, GC has been demonstrated as a field-deployable technique for the analysis of lower-volatility organics

using the Thermal desorption Aerosol Gas Chromatograph (TAG), particularly with recent work expanding its application to

oxygenates that might be detectable by iodide CIMS (Bi et al., 2021b; Isaacman-VanWertz et al., 2016; Isaacman et al.,

2014; Thompson et al., 2017; Williams et al., 2006; Zhao et al., 2013). Typically, detection of analytes eluting from a GC is

achieved by either flame ionization detector (FID), which has near-universal response but provides no chemical information

about an analyte (Grob and Barry, 2004; Kolb et al., 1977), or an electron ionization mass spectrometer (EI-MS). The latter

provides identification of compounds with mass spectra available in existing libraries, but structural or molecular

information of compounds not in those libraries requires careful interpretation of mass spectra. Unfortunately, compounds

not in existing libraries account for a substantial fraction of compounds in SOA. This shortcoming has, in part, led to recent

efforts to couple GC with CIMS for detection to provide the classification of unknown analytes by their elemental formulas

(Bi et al., 2021b; Koss et al., 2016; Vasquez et al., 2018).

We recently demonstrated a coupled TAG-CIMS/FID, in which particle-phase organics are collected and analyzed by a

TAG, with analyte detection simultaneously achieved by an FID and an iodide CIMS (Bi et al., 2021b). This approach

allows quantification of individual analytes in particle-phase organics by FID, with simultaneous classification by their

elemental formula through CIMS. In this work, we focus on quantifying the sensitivity of an iodide CIMS to different

isomers of the same elemental formula by comparing signals of a CIMS with quantification of each isomer mass based on

FID response. Specific objectives are to 1) compare the iodide CIMS sensitivity of isomers of a given elemental formula; 2)

examine the efficacy of voltage scans to predict the sensitivity of a given analyte or formula; and 3) determine the extent to

which the additional dimension of GC retention time can inform estimates of iodide CIMS sensitivity.

## 2    Instrumentation and methods

### 2.1  Instrument operation

TAG configuration. The TAG-CIMS/FID couples a GC instrument, the TAG, with two detectors, a HR-ToF-CIMS

(Aerodyne Research Inc.) using iodide as the reagent ion and an FID (Agilent Technologies). The TAG-CIMS/FID enables

online, isomer-resolved analysis of particle-phase oxygenated organics through sample collection followed by separation of

isomers by GC. Quantification relies on an FID, calibrated by automated injection of a small number of calibrants and

internal standards. Details of the instrument, sampling procedure, and chemical analysis method are described by Bi et al.

(2021b). In brief, the TAG collects aerosol samples by impaction into a passivated steel cell at a sample flow rate of 9 slpm,

typically for 5-15 minutes in this work with an equivalent volume of 45-135 L air. Liquid chemical standards are injected



into the cell through the automated liquid injection system of the TAG (Isaacman et al., 2011). Samples collected by the cell

are then transferred to the GC column through programmed thermal desorption. A polar GC column (MXT-WAX, 17 m ×

0.25 mm × 0.25 μm, Restek) wrapped on a temperature-controlled metal hub is used for the separation of oxygenated

organic compounds (50 °C to 250 °C at a rate of 10 °C /min and then held for 25 mins). Though analysis of oxygenates by

GC (in general) or TAG (specifically) typically relies on derivatization to convert difficult-to-elute polar functional groups

(e.g., hydroperoxides) into easier-to-elute groups (Isaacman et al., 2014), this approach is not employed here to minimize

chemical alterations to the functionality of the analytes reaching the detectors. The GC column effluent is split to the two

detectors, 0.7 sccm to CIMS and 0.3 sccm to FID, using a heated and passivated tee (SilcoNert 2000, SilcoTek Corp.) with

heated fused-silica transfer lines for simultaneous measurements by CIMS and FID. The detailed validation of the split ratio

is described by (Bi et al., 2021b).


CIMS configuration. The configuration of the HR-ToF-CIMS using iodide as the reagent ion is described in detail by Bi et

al. (2021b) and is operated similar to typical direct air sampling by CIMS (e.g., Isaacman-Vanwertz et al., 2018). Major

differences between the current instrumental setup and a direct-air-sampling CIMS are highlighted here. The inlet of the

CIMS is modified by adding a 225°C metal cartridge with a bore-through hole to allow the insertion of the transfer line, a

fused-silica guard column, into the ion-molecule reactor (IMR). Helium flow eluting from the GC (0.7 sccm) mixes with 2

slpm of reagent ion flow in the IMR; the ~3000x dilution of this effluent is roughly balanced by the preconcentration of

sample in the impactor cell such that the detected concentrations are similar to those expected under typical direct-air-

sampling conditions. The CIMS is operated in two modes: regular mode and voltage scanning mode, which differ in their

data acquisition rates and voltage settings. In regular mode, to obtain a smooth chromatographic peak, raw negative-ion

spectra are acquired at a rate of 4 Hz, higher than the typical data acquisition rate for laboratory studies (1 Hz, with data

typically reported as 1 min averages). Voltage scanning mode requires even higher acquisition rates and is described below.

CIMS voltage scanning. A voltage scanning mode is applied to examine the method for the prediction of analyte sensitivity

in an iodide CIMS (Iyer et al., 2016; Lopez-Hilfiker et al., 2016). By scanning the voltage difference (dV) between the

skimmer of the first quadrupole and the entrance to the second quadrupole ion guide of the mass spectrometer, the

relationship between dV and signal fraction remaining is established and can be fit by a sigmoid function described by a

maximum possible signal ($S_0$) and a signal decay rate as a function of dV. The voltage difference at which half the maximum

signal is removed (i.e., half the adducts that could be formed are de-clustered) is described by a critical parameter, $dV_{50}$. This

parameter has been shown to correlate with the binding enthalpy of the iodide-molecule adduct and the analyte sensitivity in

an iodide CIMS (Lopez-Hilfiker et al., 2016). Quantification using these relationships has been previously shown to yield

results within 60% uncertainty in total measured carbon (Isaacman-Vanwertz et al., 2018). Other researchers have applied

variations of the voltage scan method to acetate or $NH_4^+$ CIMS, such as scanning the voltage difference at seven different

sections of the mass spectrometer (Brophy and Farmer, 2016) or using the ion kinetic energy ($KE_{50}$) instead of $dV_{50}$ (Zaytsev



et al., 2019). These approaches all seek to quantify instrument response empirically through variations in the operating

conditions; for this work, we follow the original approach described by Lopez-Hilfiker et al. (2016).

No consensus currently exists on the rate at which voltages can (or should) be scanned, the number of spectra collected at each dV level, or the number or range of dV levels scanned, but previous work has demonstrated complete voltage scans on timescales of minutes (Isaacman-Vanwertz et al., 2018; Mattila et al., 2020; Zaytsev et al., 2019). This timescale is not

practical for GC applications, in which chromatographic peak widths are typically less than tens of seconds. However, it is possible to estimate the fastest possible voltage scanning rate, which is limited primarily by the time required for the signals to settle after any change in the full set of voltages, as any data collected during this time does not accurately reflect the voltage state. The instrument used in this work was able to switch and stabilize all voltages in ~100 ms. Using these conditions as an example, we develop a rough understanding of the maximum possible rate of voltage scanning. If voltages

are varied at 5 Hz (i.e., 200 ms per voltage level), and data is collected at 10 Hz (i.e., 100 ms per data point), data would alternate between single points of "transition spectra" that must be ignored, and single points of real spectra representing the new voltage level. Such an approach is impractical, however, as each level would be represented by a single spectrum, collected immediately after the transition. Instead, to achieve a reasonable and accurate relationship between the signal fraction remaining and the dV, we switch voltages at a slower rate (2.5 Hz) and acquire data at a faster rate (20 Hz). This

approach is fast enough to provide multiple spectra per dV level and multiple dV levels across a single chromatographic peak. With this acquisition rate, eight (i.e. 20 Hz/ 2.5 Hz) data points per dV were collected, with at least the first 2-3 expected to be "transition spectra" that need to be ignored; practically speaking we find that only the last 3-4 spectra are stable (typical relative standard deviation < 20%), so the first 5 spectra of each level are ignored and the signal at a given dV level is taken as the average of the final 3 spectra collected.




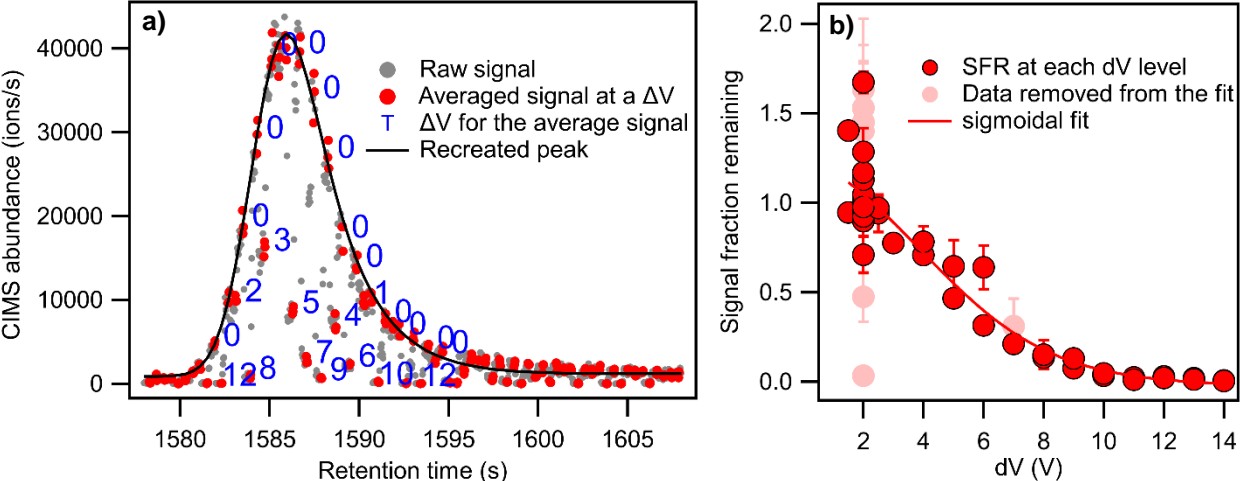

**Figure 1**. Demonstration of the voltage scan method to the GC-CIMS with a) recreation of chromatographic peak (grey dots represent all data collected at 20 Hz, while the larger red dots represent the last three data points at each dV level, which are used to calculate signal at the given ΔV level (blue text, ΔV = dV- 2 (baseline voltage)); and b) signal fraction remaining at each voltage difference (dV). Data points in light red, excluded from the sigmoidal fit, are signals from each dV level that do not meet quality control metrics.

The voltage settings of the CIMS in the regular mode are used as a set of baseline values (designated as dV=2 V). Fourteen different sets of voltage settings, each of which has a constant voltage deviation from the baseline values (ΔV = -0.5 V to +12 V). The voltage setting is varied at 2.5 Hz, alternating between the baseline values and a set of voltages representing a different dV level. The set of dV levels is always in the same order, but is not monotonic, randomized to avoid the influence of potential memory effects on the results. An example of the output data for a signal chromatographic peak is shown in Figure 1a. Grey dots represent all data collected at 20 Hz, while the larger red dots represent the last three spectra at each ΔV level (ΔV = dV − 2), which are used to calculate signal at that voltage scan with the given ΔV level (blue number, offset to the right). Signal fraction remaining (SFR) is calculated as the measured signal, S, at a certain voltage scan, *n*, of a given dV level, divided by the expected signal, S', that would have been observed at that scan using baseline voltages, *base* (here, dV=2 V):

$$SFR = \frac{S_n^{dV}}{S'_n^{base}}$$

With direct-air-sampling, signal changes sufficiently slowly that baseline signal at a given dV level can be inferred from subsequent and following measurements at the baseline voltages, i.e., $SFR = \frac{S_n^{dv}}{\frac{1}{2}(S_{n-1}^{base}+S_{n+1}^{base})}$. However, in a chromatographic application, the signal varies so rapidly due to the rise and fall of chromatographic peaks that such an approximation is not necessarily reliable. Instead, the changes of the voltage settings back-and-forth between the baseline condition and a certain voltage difference allow us to recreate the chromatographic peaks for sample runs in the voltage scan mode. As shown in Figure 1a, the signals obtained with the voltage setting at the baseline condition are used to recreate the chromatographic



peak by fitting these points with an exponentially modified Gaussian peak, recreating the peak shown in black in the example. Signal fraction remaining at each voltage level is calculated as the ratio of observed signal to the recreated peak at the same time.

An example of the obtained signal fraction remaining at different voltage differences is shown in Figure 1b. The $dV_{50}$ of the compound can be obtained by fitting the data with a sigmoidal function. Due to the fast acquisition and voltage scanning rate, some additional quality control metrics are necessary. Specifically, we reject data from a given voltage level if the relative standard deviation of the three included spectra is larger than 20%, indicating the voltage or signal level is not stable. Additionally, we observe that the setpoint voltages are not always reached as expected, so we use the reagent ion signal (which also changes with dV) to evaluate whether or not a given voltage level represents the target dV setting. Specifically, the medians of reagent ion signals at each dV throughout a run cycle are used as standard values of reagent signals per dV, and data are rejected if their corresponding reagent signals are more than 15% away from the standard value, indicating the target dV setting was not reached. Signals from each dV level that do not meet these quality control metrics (i.e., those in light red in Figure 1b) are not included in the sigmoidal fit. The $dV_{50}$ of each compound was calculated from duplicate samples, and was excluded from analysis if found to differ by more than 50%.

## 2.2 Experimental setup

The TAG-CIMS/FID is used to quantify SOA generated through gas-phase $O_3$ and/or multiple levels of OH oxidation of limonene (Sigma Aldrich, 97% purity) and 1,3,5-trimethylbenzene (TMB) (Sigma Aldrich, 98% purity) in a Potential Aerosol Mass (PAM) oxidation flow reactor (OFR) (Lambe et al., (2011). For the convenience of the discussions later, a given set of oxidation experiments is discussed as "precursor-oxidant" (e.g., limonene-$O_3$).

Experiments were conducted at 25℃, 40-50 % relative humidity, and a constant gas flow rate of 12 L min$^{-1}$ through the OFR. Limonene or TMB was injected into a carrier gas of synthetic air through use of an automated syringe pump at liquid flow rates ranging from 0.95 to 1.9 μL h$^{-1}$ or corresponding mixing ratios of 236-472 ppbv. During ozonolysis experiments, 35 ppmv $O_3$ was injected at the OFR inlet. During photooxidation experiments, OH and HO$_2$ were generated via O$_2$+ H$_2$O photolysis at 254 and 185 nm; over the range of conditions that were used, the estimated OH exposures in the OFR were in the range of $(4\times10^{10} - 7\times10^{11})$ molecules cm$^{-3}$ s$^{-1}$ (Rowe et al., 2020).

Between sampling from the flow reactor, liquid standards were introduced into the sample collection cell using the automated liquid standard injector on TAG. Standards analyzed included: 1,12-dodecanediol (Sigma Aldrich, 99% purity), vanillin (Sigma Aldrich, 99% purity), undecanoic acid (AccuStandard, 100% purity), hexadecanoic acid (AccuStandard, 100% purity), levoglucosan (Sigma Aldrich, 99% purity), and an n-alkanes mixture ($C_{10}$-$C_{40}$, AccuStandard, 50 μg/ml).



### 2.3 Data analysis

CIMS and FID chromatograms are collected as individual files in each GC run cycle. For each cycle, elemental formulas are identified through the high-resolution fitting of peaks in the mass spectra using the Tofware (Tofwerk, AG and Aerodyne Research, Inc., version 3.1.2) toolkit developed for the IGOR Pro 7 analysis software package (Wavemetrics, Inc.). The chromatograms (i.e., time-series data) of identified formulas in CIMS as well as the FID chromatograms are then imported into TERN, the freely-available (https://sites.google.com/site/terninigor/, last date of access, 05/26/2021) Igor-based software tool for the quantification of chromatographic data (Isaacman-VanWertz et al., 2017), with custom modifications to analyze voltage scanning data. Integration of CIMS peaks yields units of CIMS-response × s, where CIMS response is ions/s (i.e., counts per second, "cps") normalized to the number of reagent ions (typically in millions). CIMS peak areas are therefore in the units of (ions/million reagent ions) /s × s = ions/million reagent ions.

Quantification by FID. Analytes are quantified by integrating chromatographic peaks detected by the FID. Sensitivity of the FID to an oxygenated organic compound is estimated based on its elemental formula, identified by the CIMS, as described by Hurley et al. (2020). In brief, FID detection of hydrocarbons provides a near-universal response per unit carbon mass, which is easily obtained by a multi-point calibration to a hydrocarbon. The average response to carbon in oxygenates decreases proportionally to the oxygen-to-carbon ratio (O/C) of the compound. Though the exact decrease in response is driven by the chemical functional groups present, Hurley and co-workers have shown that per-carbon FID sensitivity can be estimated from O/C to within approximately 20% uncertainty for an individual analyte. We therefore calculate the mass or number of moles of an analyte from its FID peak area based on a calibration response factor to n-alkanes, with a correction for oxygenation based on the elemental formula identified by CIMS (specifically, the FID response per carbon atom relative to n-alkanes = -0.54 O/C + 0.99, where O/C is the oxygen to carbon ratio in the target analyte, Hurley et al. (2020)). The sensitivity of an analyte in CIMS, ions generated per mole introduced per million reagent ions, can be determined by dividing its CIMS peak area (ions/million reagent ions) by the number of moles calculated based on the FID peak, yielding units of ions/mole/million reagent ions. Due to the peak integration, this unit is atypical in the CIMS scientific community. Therefore, we also provide a conversion of this unit to the more common metric of CIMS sensitivity, cps/ppt/million reagent ions, based on a specific CIMS operating condition (i.e., 100 mbar in IMR; 2 slpm sample flow rate; and 2 slpm reagent ion flow rate). The detailed method of unit conversion is described in the supporting information, but essentially involves a conversion by the number of moles entering the instrument per time for a given ppt and flow rate. In the analyses presented here, the sensitivity of an analyte is calculated in three different samples to avoid potential artifacts from data analysis; these three samples represent triplicate samples for the limonene-$O_3$ experiment, and three different OH oxidation levels for the precursor-OH experiments. To avoid potential errors introduced by poor chromatographic resolution, low signal, or other issues, we exclude from the analysis any analytes for which the sensitivity calculated from the three samples have a relative standard deviation greater than 50%. It is critical to note that, because the FID is a single-channel detector, quantification by





FID is only possible for peaks that are sufficiently well-resolved to be confidently integrated and is not available for every
peak observed by CIMS.

Since the determination of analyte sensitivities relies on both CIMS and FID peak area, it is crucial to make sure that the
peaks of analyte from CIMS and FID are correctly aligned. To align peaks of the same compounds between CIMS and FID
in chromatograms, retention times are corrected based on the linear regression of retention times of internal standards (i.e.,
vanillin and 1,12-dodecandiol) as well as the two largest peaks. We reject compounds that have differences in peak retention
time between CIMS and FID more than 2 seconds after the retention time correction.We also reject compounds that have
significant differences in peak shapes between CIMS and FID.

Instrument blank runs (i.e., runs without sample collection and liquid standard injections) are conducted prior to each
oxidation experiment to make sure that there are no visible chromatographic peaks that can interfere with the data analysis
later. Blank runs are also tested every five sample runs to check for carry-over or residuals of compounds within the
instrument and no carry-overs of analytes are detected. Additionally, triplicate sample runs are conducted to test for the
stability and repeatability of the TAG-CIMS/FID. The results suggest that the relative standard deviation of signals in
triplicate runs are within 15% for CIMS and FID.




# 3    Results and discussion

## 3.1    Variability in isomer sensitivity

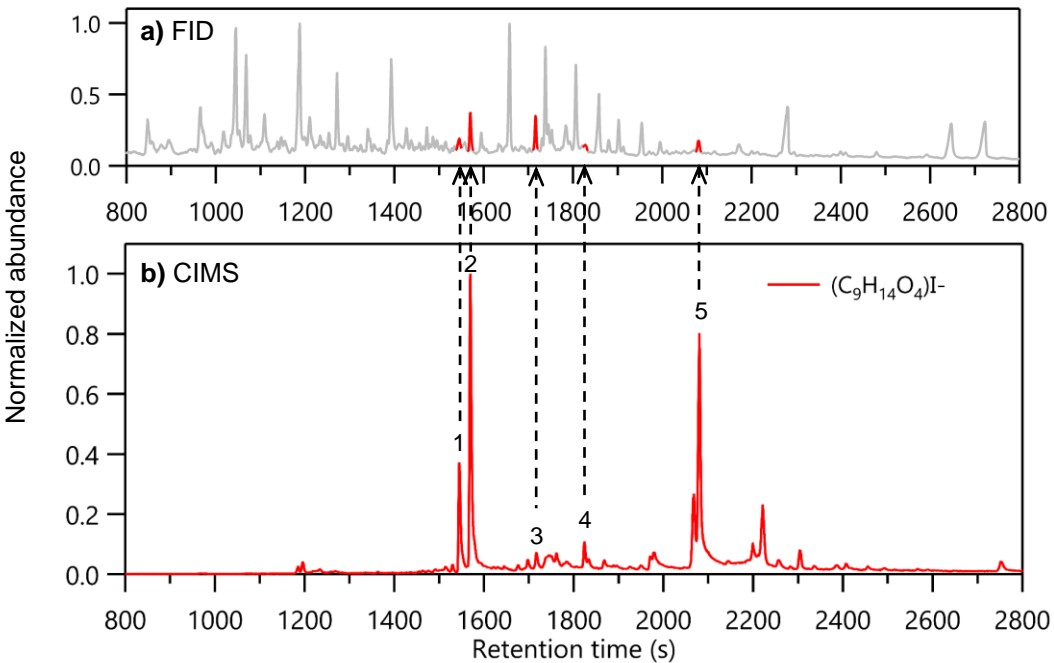

**Figure 2.** An example of isomers quantified in b) the chromatogram of ion $(C_9H_{14}O_4)I^-$ in CIMS with a) their corresponding FID peaks (highlighted in red). The peaks marked with numbers are those included in the analysis of isomer sensitivity.

As an example of the data, Figure 2b shows the chromatograms of the ion $(C_9H_{14}O_4)I^-$ from a sample of aerosol collected in the limonene-OH experiment. The FID abundance (y-axis of Figure 2a is a single-channel signal of total ions produced by the mass of carbon combusted (Holm, 1999), so each analyte (i.e., chromatographic peak eluting at a given retention time) responds with similar mass-based sensitivity. Co-eluting peaks are not well resolved and may not be able to be accurately integrated, as there is no additional dimension of separation (e.g., mass spectra) to improve resolution beyond what is shown. In contrast, CIMS signals include separation by mass of detected ions, so co-eluting analytes of different elemental formulas can be easily resolved. The chromatogram displayed, Figure 2b, is the normalized ion count signals of a single ion, $(C_9H_{14}O_4)I^-$. Five isomers, highlighted with numbers in Figure 2b, were able to be matched to FID peaks to calculate sensitivities with relative standard deviations less than 50%. While larger peaks can be easily correlated between the FID and CIMS by retention time, compounds with small peak areas such as Compound 3 and 4 in Figure 2b are also correlated when their peak shapes between FID and CIMS are similar and they have comparable behavior across different oxidation levels to ensure the proper peak assignment. As an example, Figure S1 shows the peaks representing Compounds 3 and 4, which have the same retention time and peak shape in the CIMS and FID and follow the similar trends with the change of OH exposure levels.



Given that the FID sensitivity of oxygenated organics is primarily a function of carbon and oxygen content (to within ~20%) (Hurley et al., 2020), and these isomers all have the same elemental formula, FID peak areas (highlighted in red in Figure 2a)

are proportional to the number of moles of those compounds. Differences in sensitivity are qualitatively clear, for example, Compound 2 and 3 have relatively similar number of moles in the sample (i.e., similar FID areas) yet show at least one order of magnitude difference in CIMS response. The CIMS sensitivities of the five compounds can be quantified as discussed, and are found to span the range of: $3.1 \times 10^{16}$ ions/mole/million reagent ions (Compound 1) to $6.5 \times 10^{14}$ ions/mole/million reagent ions (Compound 3). The two orders of magnitude range of sensitivities of the five identified isomers show that

although sharing the same elemental formula, isomers may have significantly different CIMS sensitivities. These differences may result in biases during quantification when using a direct-air-sampling CIMS without GC pre-separation that provides resolution of isomers.

Calibration of CIMS using FID requires the target analyte to be detected by both detectors and have a well-resolved

chromatographic peak in FID. For example, there are certainly other isomers (i.e., chromatographic peaks) in Figure 2b besides the highlighted five ones. However, some of those isomers are not included in the discussion because no FID peaks or well-resolved FID peaks are present at the same retention time as their CIMS peaks. Conversely, it is possible that some of the FID peaks are isomers of this formula that are not detectable by CIMS. This limitation can be mitigated by collecting data under a wide range of conditions or environments. Once the CIMS sensitivity of a compound is obtained, quantification

can be achieved for those compounds in other poor-signal conditions or even without the coupling of the FID.

Additionally, the use of a GC column, which is selective towards a certain range of volatility and polarity of compounds, limits the detection to specific ranges of compounds, and/or could induce thermal decomposition of some sampled compounds to form analytes not present in the original sample (Isaacman-VanWertz et al., 2016). Nevertheless,

quantification of individual analytes can be critical for understanding source and chemical pathways (Nozière et al., 2015), and can provide fundamental insight into the capabilities and limitations of a given ion reagent chemistry. Furthermore, while decomposition during analysis may impact the scientific interpretation of the collected sample, decomposition is expected to primarily occur during desorption or GC analysis and is therefore upstream of the detectors; both detectors consequently "see" the same analyte whether or not decomposition occurs, and it does not impact measurements of CIMS

sensitivity of the molecules that do reach the detectors.





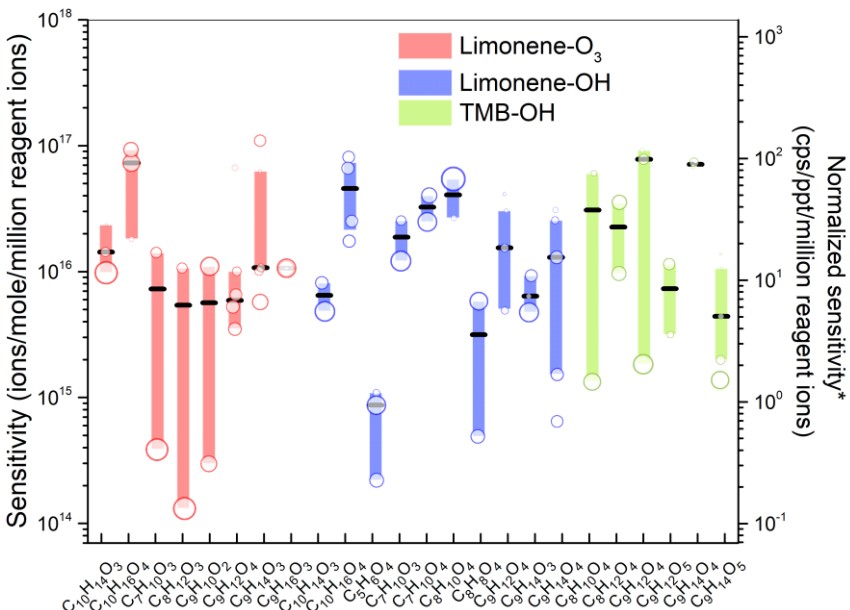

**Figure 3.** Sensitivities of constituent isomers of formulas for which at least two isomers had sensitivities obtained in the oxidation experiments. Each circle shows the sensitivity of an isomer and the area of circle represents the mole fraction of the isomer in the formula. Box represents the first to the third quartile. Black lines are the median values of the sensitivities.
*: unit converted for direct-air-sampling CIMS using 100 mbar in IMR, 2 slpm sample flow rate, and 2 slpm reagent ion flow rate.

To systematically study the variance of isomer sensitivity, formulas with multiple isomers identified in the oxidation experiments are summarized in Figure 3 (as noted above, only isomers having sensitivities with less than 50% relative standard deviation in three samples are included). The results suggest that the sensitivity of isomers typically vary by one

order of magnitude with a maximum deviation of two orders of magnitude, for instance, in the case of $(C_8H_{12}O_3)I^-$ in limonene-$O_3$, $(C_9H_{14}O_4)I^-$ in limonene-OH (also shown in Figure 2), and $(C_9H_{12}O_4)I^-$ in TMB-OH. In a minority of cases, sensitivities vary by only a factor of two to four (e.g., $(C_9H_{16}O_3)I^-$ in limonene-$O_3$, $(C_{10}H_{14}O_3)I^-$ in limonene-OH, and $(C_9H_{14}O_4)I^-$ in TMB-OH). Notably, molecules of the same formula produced through two different chemistries (e.g., $(C_9H_{14}O_4)I^-$ in limonene-OH and TMB-OH) also differ to approximately the same degree, supporting the conclusion that a

measured formula may consist of a different set of isomers depending on the sampling environment. The significant variance of isomer sensitivity indicates that if a CIMS with direct air sampling is used, the concentration of a given formula may be significantly biased towards the concentration of the most sensitive isomer within the formula while other isomers, which could be actually more abundant on a per mole basis, may be overwhelmed. As an example, consider $(C_8H_{12}O_3)I^-$ in the limonene-$O_3$ experiment, which includes a low-sensitivity, high-concentration isomer, and a high-sensitivity, low-

concentration isomer. The low-concentration isomer is approximately 80x more sensitive, but 5x less abundant (represented by the ratio of the marker area in Figure 3), than the high-concentration isomer. In this example, ~95% of CIMS signal is from an isomer that counts for less than 20% of the mass, which could introduce biases in data interpretation (if, for instance,



the two isomers can come from different sources, or represented different chemistries). The example demonstrates that isomer resolution, achieved by coupling a CIMS with a GC and a FID, consequently provides not only additional detail about a sample, but is also critical for the quantification of less sensitive isomers.

## 3.2 Prediction of sensitivity using $dV_{50}$

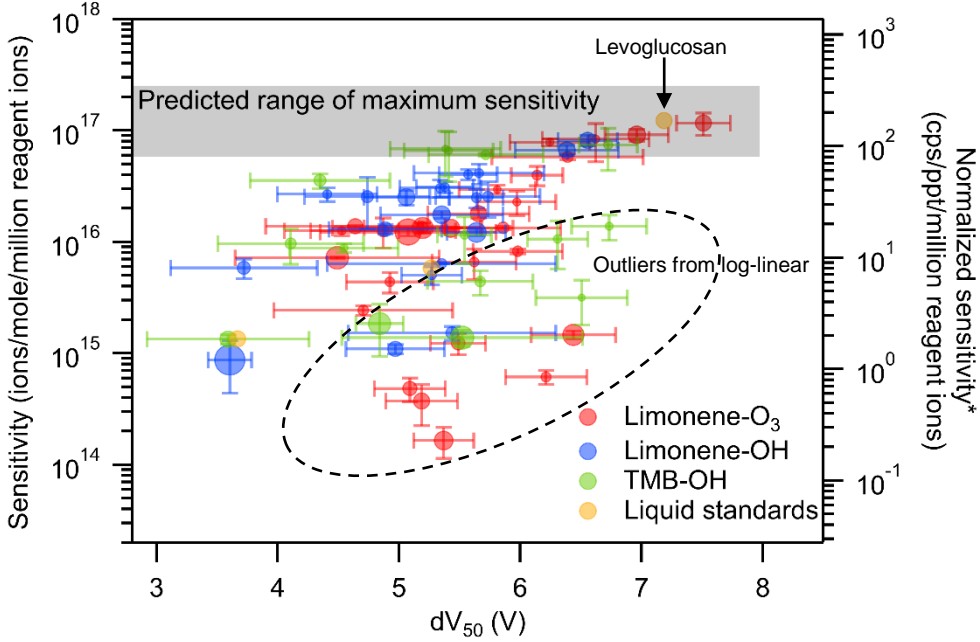

**Figure 4.** Relationship between sensitivities and $dV_{50}$ of compounds identified in oxidation experiments as well as liquid standards. Each data point is a compound identified with the marker area representing the moles of the compound. The error bars in y-axis are the standard deviation of sensitivity in triplicate (Limonene-$O_3$) or three different OH level measurements (Limonene-OH and TMB-OH). The error bars in x-axis are the standard deviation of $dV_{50}$ in duplicate measurements.
*: unit converted for direct-air-sampling CIMS using 100 mbar in IMR, 2 slpm sample flow rate, and 2 slpm reagent ion flow rate.

Previous work has shown a log-linear relationship between sensitivity and $dV_{50}$. However, these relationships were established based on only a limited number of chemical standards, mostly mono- and di-acids (Brophy and Farmer, 2016; Iyer et al., 2016; Lopez-Hilfiker et al., 2016), and demonstrate significant scatter around the trend. In this work, the quantification of compounds based on FID response allows us to broaden the investigation of this relationship from liquid standards to oxidation products. Of all observed analytes, sensitivities calculated for a total of 63 oxidation products and 3 liquid chemical standards passed all the data quality checks for inclusion in this analysis (i.e., sufficient FID resolution for quantification, relative standard deviations of sensitivity less than 50% for triplicate (Limonene-$O_3$) or three different OH level measurements, relative standard deviations of $dV_{50}$ less than 50% for duplicate measurements). The calculated sensitivities of all 66 compounds and their $dV_{50}$ obtained using the voltage scan method are plotted in Figure 4. Sensitivities of analytes included in the discussion vary across three orders of magnitude. A plateau of sensitivity, which is an indication of maximum sensitivity, is observed for compounds with dV greater than 7 V. Levoglucosan, which is known to be detected



at the collision limit (Lopez-Hilfiker et al., 2016), is one of the compounds that nearly reach the maximum sensitivity in this study. The finding agrees with those reported in the literature that the maximum sensitivity can be reached if the iodide-molecule reaction is only limited by the formation rate (i.e., collision limit) (Huey et al., 1995).


The observed plateau of sensitivity is in the range expected for maximum sensitivity (calculated in the supporting information) based on instrument operating conditions; the colored gray bar in Figure 4 spans the range from the calculated kinetic-limited maximum sensitivity to 4 times lower values (observed by Isaacman-VanWertz et al. (2018) to be the maximum sensitivity using an instrument of the same design as that used here). The right axis provides a direct mathematical conversion between the left axis and more typical CIMS units assuming a sample flow of 2 slpm (as opposed to the 0.7 sccm used here). It provides general context for the conversion between the units, but is not fully representative of the conversion due to differences in flow between our setup and typical operation and its concomitant impacts on the residence time within the reaction region. While the detailed analysis is in the supporting information, true max sensitivity under our IMR conditions but with typical flows is found to be 88 cps/ppt/million reagent ions (not 350 as implied in Figure 4). For this large set of individual analytes, the relationship between sensitivity and $dV_{50}$ shown by Iyer et al. (2016) and Lopez-Hilfiker et al. (2016), is not so clear. A log-linear relationship defines an apparent upper bound of sensitivity, but approximately one-quarter of compounds (dashed region) have sensitivities substantially lower than this relationship. These results suggest that for calibrating individual components, the log-linear relationship between sensitivity and $dV_{50}$ may provide some rough indications of sensitivity but is fairly imprecise.




The data shown in Figure 4 is inherently different than the data from a typical, direct-air-sampling application of CIMS in ways that could impact the application of the relationship between sensitivity and $dV_{50}$. Direct-air-sampling CIMS classifies analytes by elemental formula basis while a TAG-CIMS can differentiate isomers and provide quantification down to the isomer-resolution. The analytes found to be outliers of the log-linear relationship are mostly less sensitive compounds in CIMS so their responses might be overwhelmed by the signals of more sensitive isomers with the same formula. Such an outcome would potentially strengthen the observed log-linear relationship but would underestimate the observed mass of that compound (and consequently formula). To examine the applicability of the voltage scan method to a CIMS operated with direct air sampling, we convolute resolved isomers into their elemental formulas. The sensitivity of a formula is calculated as the average of isomer sensitivities weighted by their number of moles, while the $dV_{50}$ of a formula is calculated as the average of $dV_{50}$ weighted by their CIMS abundance (i.e., chromatographic peak area in CIMS data). The result of averaging and weighting each parameter in these ways recreates how that formula would respond in direct-air-sampling CIMS if it had the same isomer composition; we note that a direct comparison cannot be made by a simple direct air sample of the same mixture due to artifacts introduced by the GC (both positive - the formation of new compounds through thermal decomposition, and negative – the inability to elute highly polar compounds). To ensure averaged formulas are a reasonable representation of a hypothetical direct-air CIMS sample, formulas are removed from the analysis if (1) the formula has only






two isomers and one of the isomers does not have a calculated sensitivity; and/or (2) the isomer with most abundant CIMS signal does not have a reported sensitivity. The resulting relationship between sensitivity and $dV_{50}$ on a per formula basis can be plotted (Figure 5). The compounds that are filtered by these criteria are not statistically significantly different from the compounds that pass these criteria (one-way ANOVA test, p=0.36 and 0.71 for log-transformed sensitivity and $dV_{50}$, respectively).

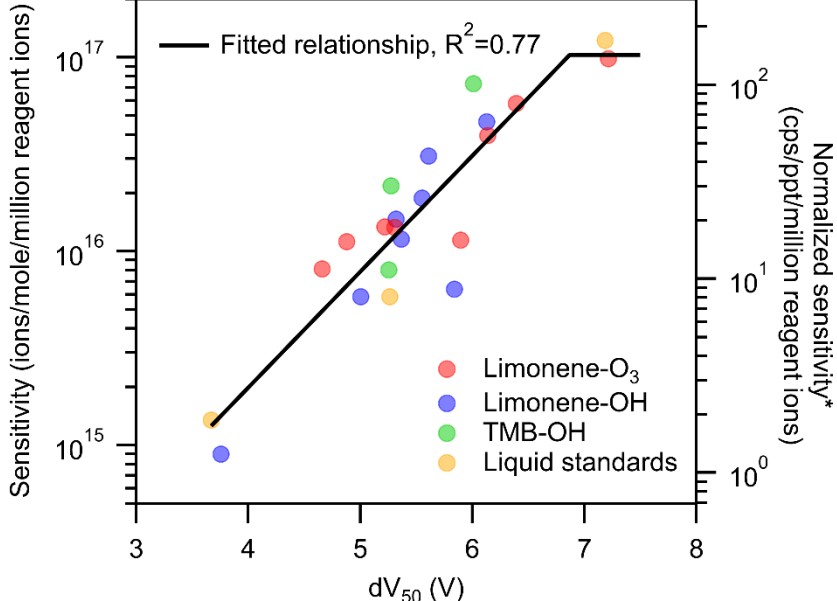

**Figure 5.** Sensitivity vs $dV_{50}$ on a per formula basis with a linear regression on the formulas.
*: unit converted for direct-air-sampling CIMS using 100 mbar in IMR, 2 slpm sample flow rate, and 2 slpm reagent ion flow rate.

As shown in Figure 5, the log-linear relationship improves when considered on a per formula basis, though significant scatter remains. Linear regression for formulas of $dV_{50} < 7$ V (i.e., having lower than maximum sensitivity) has a reasonable correlation ($R^2$=0.77) with a decrease of 0.6 log units (i.e., a factor of $10^{0.6}$) of sensitivity per volt change in $dV_{50}$. This slope is similar to, but slightly shallower than, the slope observed in previous work of 0.9 log units per volt (Lopez-Hilfiker et al., 2016).

The goal of the voltage scan calibration approach is to predict the sensitivities of analytes without conducting individual calibrations with chemical standards in permeation tubes. To quantify the error introduced in this approach, we use the fitted linear regression equation in Figure 5 to calculate a sensitivity for each compound based on its observed $dV_{50}$. As described by Bi et al. (2021a), sensitivity predicted in log-linear based calibration method is inherently biased and the bias can be corrected based on residual scatter around the nominal relationship, $\sigma_{scatter}^{eff}$; given the calculated $\sigma_{scatter}^{eff}$ of residual is 0.22, predicted sensitivities are expected to by biased low by a factor of 1.14, so are adjusted by that factor here. In Figure 6a, the



moles of each compound calculated using the fit from their $dV_{50}$ ("fitted moles") is compared to its measured moles
calculated the FID data. Compounds that were filtered out of Figure 5 using the criteria described above are included in the
results shown in Figure 6. By including compounds not used to generate the linear relationship, this approach therefore
provides a more realistic and conservative evaluation of the approach. The results show that about 60% and 80% of
compounds are estimated within a factor of 3 and 10 uncertainties, respectively, indicating that the voltage scan approach has
high uncertainties for individual components. Additionally, to examine the accuracy of the predictions for a direct-air-
sampling CIMS, we compare the fitted moles with the measured moles on a per formula basis (i.e., the summation of moles
of isomers within each formula) in Figure 6b. Most of the formulas are estimated within a factor of 3 uncertainties. Notably,
a factor of 3 is in approximate agreement with the uncertainty previously estimated for individual components quantified by
iodide CIMS voltage scanning (Isaacman-Vanwertz et al., 2018).

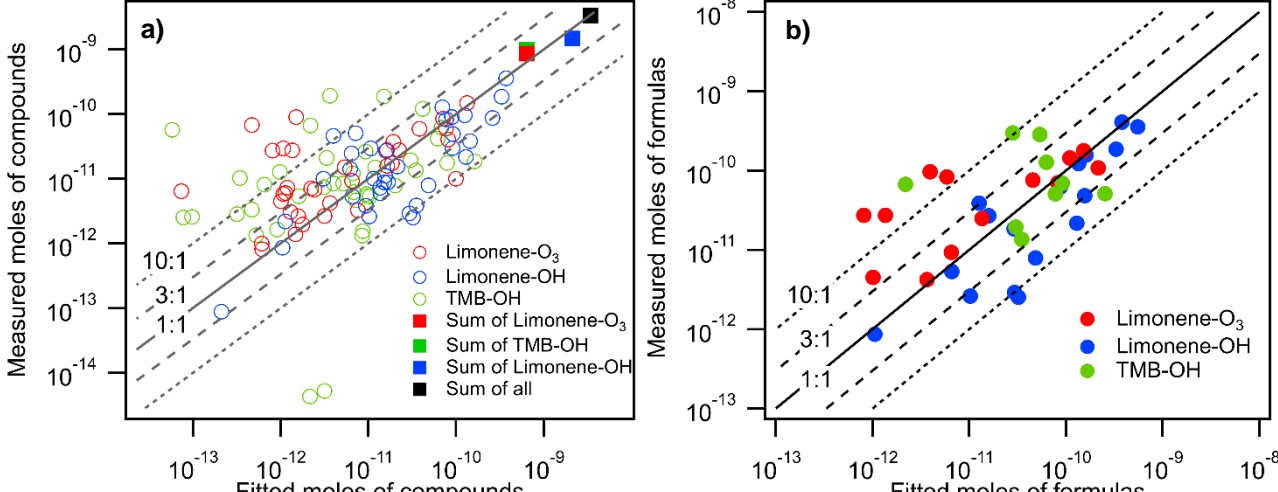

**Figure 6.** Measured moles of a) compounds and b) formulas (i.e., summation of isomers within each formula) in all oxidation experiments vs their fitted moles using the linear regression equation obtained in Figure 5.

Because the log-linear relationship of voltage scanning provides a reasonable central tendency, its application does not
introduce significant bias overall. The predicted total moles of compounds measured agree well with the measured total
moles (shown as rectangular markers in Figure 6a, with errors within 30% within an oxidation system and for all oxidation
systems combined), although predicted moles of individual compound have relatively high errors. The finding agrees with
earlier statistical analysis suggesting that the summation of multiple analytes with high scatters of sensitivity around a
nominal relationship can reduce uncertainty (Bi et al., 2021a), and is in qualitative agreement with the finding by Isaacman-
VanWertz et al. (2018) that uncertainty in the sum of ions was substantially lower than uncertainty in an individual analyte.
We conclude that, although using voltage scanning introduces high error into the estimation of sensitivity for individual
compounds, the approach provides a reasonable estimate of the summed total abundance.



## 3.3 Implications of retention time index

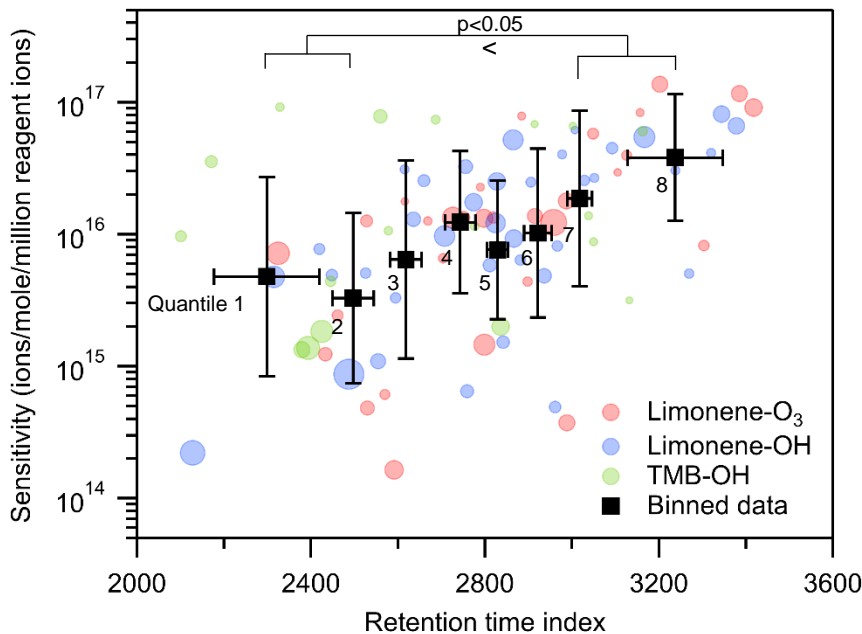

**Figure 7.** Relationship between sensitivities and retention time index of all compounds in oxidation experiments. Black markers are all data equally divided into eight bins based on the ranking of their sensitivity and retention time index, centered on averages with error bars representing the standard deviation of sensitivity and retention time index. The size of the round marker represents the number of moles of each compound.

The use of a GC expands the data with another dimension, retention time, which is generally governed by the polarity and vapor pressure of the compound and could potentially provide additional information to inform estimation of CIMS sensitivity. Since a GC column with a polar stationary phase is used in the TAG-CIMS/FID, we expect the compound with high polarity and/or low vapor pressure to have larger retention time in the chromatogram. Polarity and vapor pressure are not fully independent: the presence of polar functional groups (e.g., hydroxyl and carboxyl groups) tends to accompany

larger decreases in vapor pressure than less polar groups (e.g., carbonyl groups) (Kroll and Seinfeld, 2008). Retention time is therefore expected to also positively correlate with iodide CIMS sensitivity, which generally increases with the presence of polar functional groups (Lee et al., 2014). To examine the hypothesis, we plot the relationship between sensitivity and retention index for compounds identified in oxidation experiments in Figure 7 (retention index = retention time of an analyte adjusted such that *n*-alkanes are defined to elute at spacings of 100 units (Onuska and Karasek, 1984)). The results suggest

that there is a qualitative linear trend between log(sensitivity) and retention time index, particularly for analytes with higher abundances. Binning the data into equally distributed groups (octiles) reveals the tendency for later retention times to accompany higher sensitivity. Specifically, the latest eluting two bins are more sensitive than the earliest eluting two bins with statistical significance of $p < 0.05$ using a Wilcoxon signed-rank test; differences between intermediate bins are not statistically significant. A major driver of retention time is molecule size (e.g., number of carbon atoms), which does not





necessarily impact iodide CIMS sensitivity. Consequently, the coarseness and scatter of the relationship between retention

time and iodide CIMS sensitivity may be due to the concurrent impacts on retention time of polarity and vapor pressure.

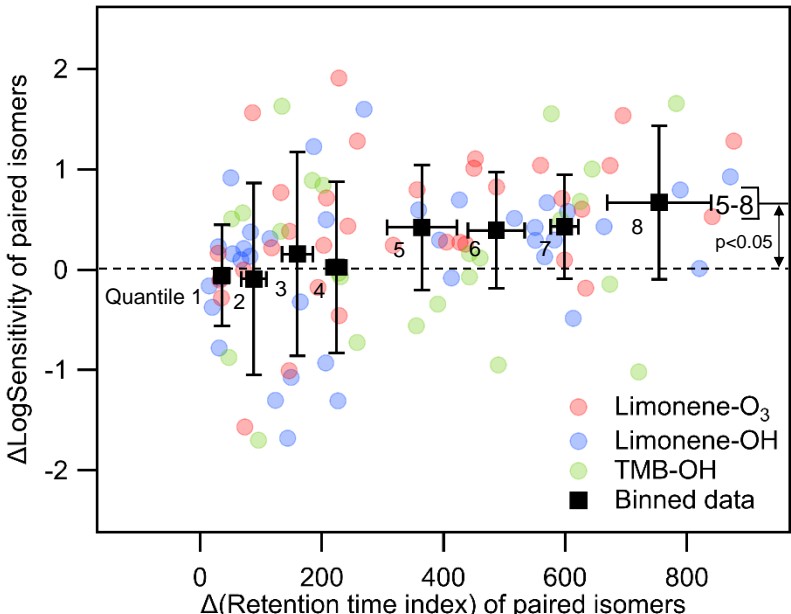

**Figure 8.** The relationship between differences (Δ) in log(sensitivity) and retention index for all pairs of isomers with a given formula. Black markers are all data equally divided into eight bins (octiles), centered on averages with error bars representing standard deviations.

To better isolate the effects of polarity in the relationship between retention time and vapor pressure, we attempt to constrain

molecule size. By examining differences between retention times of compounds with a given elemental formula, the effects

of molecular structure and differences in chemical functionality can be separated from some of the features that impact vapor

pressure (number of carbon atoms, molecular weight, etc.). If two isomers have similar chemical structure and functional

group, their retention time should be relatively close due to the minor differences in vapor pressure and polarity, and their

iodide CIMS sensitivity is likely to be roughly similar. Conversely, an isomer with a higher retention time index is expected

to contain more polar functional groups, which are also expected to have a large impact on vapor pressure. To test this

hypothesis, we compare the sensitivities and retention index of all isomer pairs (i.e., for a formula of $n$ isomers, there are

$n*(n-1)/2$ unique isomer pairs). For each isomer pair, we calculate the difference between the log(sensitivity) of the later-

versus earlier-eluting isomer and compare to their difference in retention index, shown in Figure 8. Eight equally distributed

bins (octiles) are included in Figure 8 to better identify trends and allow statistical comparisons. Qualitatively, it is apparent

that isomers with small differences in retention time vary widely in their sensitivities, frequently differing by one order of

magnitude, while later-eluting isomers tend to have higher sensitivities. Isomers that elute substantially later (octiles 5-8)

have statistically higher sensitivities by, on average, roughly by half an order of magnitude (a factor of 3 to 4). Conversely,

there is no statistical difference in the sensitivities of isomers with retention indices within ~300 of each other (octiles 1-4).



In other words, while iodide CIMS sensitivity of isomers with similar retention time is not a strong function of retention time, later elution does indicate some tendency for higher sensitivity, presumably driven by the presence of higher polarity functional groups.

While the analysis here does not provide a sufficiently deep understanding of the relationship between sensitivity and column retention to produce a quantitative approach for estimating sensitivity, it does demonstrate an approach by which interactions with the GC column can provide insight into iodide CIMS sensitivity. These data suggest that the properties of a molecule that drive iodide CIMS sensitivity are correlated, but not tightly, with the properties that drive retention time of this particular GC stationary phase. Future detailed studies and physicochemical modeling of the column retention of analytes could make use of these relationships to better understand the factors driving sensitivity and selectivity of a given CIMS reagent ion chemistry.

## 4    Conclusions

By coupling a TAG simultaneously to a FID and an iodide CIMS, we quantify isomer-resolved CIMS sensitivity (i.e., CIMS signal divided by the number of moles of analytes quantified using FID signal) for liquid standards as well as oxidation products for which commercial chemical standards are not available. The variance of isomer sensitivities for oxidation products in an iodide CIMS is found to be generally one order of magnitude and up to two orders of magnitude. The wide range of isomer sensitivities indicates that if an iodide CIMS with direct air sampling is used, measurements of formulas are likely to minimize the contributions of certain (likely less polar) isomers. The concentration of a given formula would be expected to be biased towards the concentration of the most sensitive isomer within the formula, even in cases where this isomer is not the most abundant.

We then investigate the previously reported log-linear calibration relationship between iodide CIMS sensitivity and $dV_{50}$ by applying the relationship to a broader range of chemicals including oxidation products. We find that estimating sensitivity of a given compound from its declustering voltage (i.e., $dV_{50}$) carries high uncertainties (half to one order of magnitude). However, the voltage scan approach to calibration can be used to estimate aggregate/summed moles of all analytes with low error (~30% differences between predicted and measured moles) since summing multiple analytes statistically reduced the uncertainty in the sum (Bi et al., 2021a). These results imply that in the interpretation of direct-air-sampling CIMS data, quantification based on declustering voltage is highly uncertain for individual compounds and relatively uncertain for individual formulas. Nevertheless, a nominal voltage scanning relationship built using elemental formulas represents a central tendency and can be used to estimate total mass or moles reasonably well.



We further find that the additional dimension of GC retention time provides some possible advantages to understand iodide

CIMS sensitivity. There exists a positive relationship between retention time on the GC column and iodide CIMS sensitivity, but the relationship is not yet sufficiently well understand to become quantitativey useful. Future work is needed to investigate the relationship between GC retention time and iodide CIMS sensitivity.

**Data availability**

All raw and processed data collected as part of this project are available upon request.

**Author contributions**

CB led the instrumentation design, experimental setup, and the consequent data analysis under the guidance of GIVW. GIVW, JEK, BML, and MRC contributed to the development of the theory of the described approach. GIVW, GOF, JEK, JTJ, and DRW contributed to hardware design and instrumentation. JEK, WX, ATL, MSC contributed to data collection and analysis. CB prepared the manuscript with contributions from all authors.

**Competing interests**

JEK, MSC, WX, ATL, JTJ, DRW, BML, and MRC are employed by Aerodyne Research, Inc., which commercializes TAG and CIMS instruments for geoscience research.

**Acknowledgments**

This work is primarily supported by the Alfred P. Sloan Foundation Chemistry of the Indoor Environment Program (P-2018-

515 11129).

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
