# Peer review of "Quantification of Isomer-Resolved Iodide CIMS Sensitivity and Uncertainty Using a Voltage Scanning Approach"

_Atmospheric Measurement Techniques, 2021_

## Author Comment (AC1)

The authors would like to thank the reviewers for the feedback on the manuscript. We have made revisions to the manuscript according to the reviewers' comments and the extra experimental findings. The colorings of text in the reviewer response are:

- Light blue: Original reviewer comments
- Dark blue: Text added in the revision while strikethrough words are the text deleted in the revised manuscript.
- Black: Original text in the submitted version of the manuscript and authors' response to the comments and others.

Note that the line number in the response is based on the revised clean-version manuscript.

**Reviewer 1:**

The authors investigated the calibration method for the Iodide CIMS with a Thermal desorption aerosol gas chromatograph (TAG) and an FID detector, allowing the determination of isomer-resolved sensitivity. For the same formula, sensitivities for different isomers were found to vary by 1-2 orders of magnitude. The results suggested calibration based on direct air sampling can be biased towards isomers with higher sensitivity. Sensitivity estimation using voltage scanning method after GC separation was compared to direct calibration (without column separation) and showed a high uncertainty by 0.5-1 order of magnitude. They also found that iodide CIMS sensitivity correlates with GC retention time, however, more work is needed for a calibration purpose.

This paper is well written with informative description. Results of the study can be useful for future applications of similar method. I have a few specific comments.

**Specific comments:**

Comment 1: This study measured particle phase OH and ozone oxidation products. These products are likely heavier and more oxidized/functionalized (thus less volatile) than gas phase compounds. Can the authors comment on the applicability of these calibration techniques to gas phase measurements, especially CIMS is often used for gas phase measurement? Would they expect similar results? Are there any suggestions for future applications?

Response: We would like to thank the reviewer first for thinking this manuscript useful for future applications. We believe that the calibration techniques (both the FID calibration approach for CIMS and the voltage scanning approach) can be applied to gas-phase measurements with some limitations.

Calibrating CIMS using FID relies on the GC separation of the analytes since FID itself cannot resolve individual chemical species from a mixture. Therefore, as long as the analyte, no matter it is in the gas- or phase-phase, can be collected by the upstream GC, separated by the GC column, and transferred to both detectors, its CIMS sensitivity can be quantified by the technique. While the thermal desorption aerosol gas chromatograph (TAG) was used for sample collection and GC separation in this work, any GC-based instrument can be used to apply this technique and the design and configurations of the GC will depend on the specific applications of the user. In this particular GC, TAG, was configured with an impactor cell that collects mostly particle-phase samples in air. However, previous work has shown that a filter cell can be used, which allows simultaneous collections of particle- and gas-phase samples (Isaacman et al., 2014). Additionally, recently developments on coupled GC-proton transfer reaction (PTR)-MS have shown that coupling a CIMS to a GC can analyze gas-phase analytes with the resolution of individual molecules (Claflin et al., 2021). Although it is not an iodide CIMS, it shows the potential to analyze gas-phase components in similar cases.

To strengthen these suggestions for future applications, we have revised the manuscript on Line 319:

"Notably, although this instrument TAG-CIMS/FID is specifically configured to collect only particlephase samples, this calibration technique for quantifying CIMS sensitivity could be applied to gas-phase samples using a different instrument configuration and should be applicable to any GC-based instrument employed upstream of the two detectors as long as the analyte can be collected by the upstream instrument, separated by the GC column, and transferred to both detectors."

As for the voltage scanning techniques, it can be applied to a CIMS with direct air sampling as demonstrated by Isaacman-Vanwertz et al. (2018) and Mattila et al. (2020). The work in this study is to quantify the uncertainty for this calibration technique. Therefore, this work will not limit its application to particle-phase only scenarios.

**Comment 2: Can the authors add some discussion on the influence of RH on the different calibration methods? It is known that sensitivity is RH dependent for I-CIMS for many chemicals.**

Response: We agree with the reviewer and have added a discussion of its influence in the context of a TAG-CIMS/FID. In brief, the relative humidity (RH) influence is minimal in coupling a GC to a CIMS, which is one merit of the coupled GC-CIMS but was not mentioned in the submitted version of the manuscript. Water vapor is not collected by the TAG impactor, and any residual moisture is purged before transfering analytes to the GC column. Analytes are transferred to the CIMS using ~1 sccm of carrier gas (i.e., helium), and this GC flow is what the CIMS samples, so the RH of the CIMS samples flow is always the same (~0%). RH consequently will not impact the CIMS sensitivities of analytes quantified by the FID calibration method. Since the TAG does not introduce moisture to the ion-molecular reaction region (IMR) of the CIMS, the RH in the IMR can be controlled at a fixed level by adding water vapor in the ~2 lpm reagent ion flow. A benefit of the proposed approach is thus that future work could be done to quantitatively probe sensitivity dependencies of RH using the coupled TAG-CIMS/FID by feeding the IMR with reagent ion flow containing different RH levels. To highlight this merit of TAG-CIMS/FID, we have revised the manuscript on Line 346:

"It is well-established that sensitivity is humidity-dependent for many chemicals in an iodide CIMS (Lee et al., 2014). Because analytes entering the IMR come from the GC in a dry helium flow, the relative humidity in the IMR is stable and can be controlled by adjusting the mixed water vapor in the reagent ion flow. Therefore, the coupled TAG-CIMS/FID provides opportunities for future work to quantitatively investigate the humidity dependency of sensitivity of those chemicals."

**Comment 3: Line 142: "similar" should be "similarly".**

**Response: We have revised the manuscript on Line 142**

"...is operated similarsimilarly to typical direct air sampling by CIMS..."

Comment 4: Figure 1: This figure is a little complicated to understand. The authors have very thorough method description on line 190-205. However, for audience that are not very familiar with the technique and the concept, it is overly technical, and they may get lost through the text. One suggestion for the authors is to make a plot showing the technique, especially the voltage scanning method coupled to GC separation, and the corresponding data collection and quantification that were used for constructing Figure 1. This can be a cartoon/plot illustration in the supporting information.

Response: We apologize for making a figure too hard to understand. We have also revised the discussions associated with Figure 1 to improve the clarity of the manuscript on Line 174:

"This timescale is not practical for GC applications, in which chromatographic peak widths are typically less than tens of seconds. However, it is possible to estimate the fastest possible voltage scanning rate, which is limited primarily by the time required for the signals to settle after any change in the full set of voltages, as any data collected during this time does not accurately reflect the voltage state. In this work, The instrument used in this work was able to switch and stabilize all voltages in ~100 ms. Using these conditions as an example, we develop a rough understanding of the maximum possible rate of voltage scanning. If voltages are varied at 5 Hz (i.e., 200 ms per voltage level), and data is collected at 10 Hz (i.e., 100 ms per data point), data would alternate between single points of "transition spectra" that must be ignored, and single points of real spectra representing the new voltage level. Such an approach is impractical, however, as each level would be represented by a single spectrum, collected immediately after the transition. Instead, to achieve a reasonable and accurate relationship between the signal fraction remaining and the dV, we switch voltages at a slower rate (2.5 Hz) and acquire data at a faster rate (20 HzGenerally, the voltages need to be switched fast enough to have multiple dV levels across one chromatographic peak, but slow enough to reach a steady state before switching again. Data acquisition must occur faster than voltage switching, with sufficient time resolution to discard any data collected during voltage transitions (i.e., non-steady-state). In this study, these requirements were met by switching voltages at 2.5 Hz and acquiring data at 20 Hz. With this acquisition rate, eight (i.e. 20 Hz/2.5 Hz) data points per dV were collected, with at least the first 2-3 as "transition spectra" that need to be ignored; practically speaking we find that only the last 3-4 spectra are stable (typical relative standard deviation

**Figure 1**. Demonstration of the voltage scan method to the GC-CIMS with a) recreation of chromatographic peak (grey dots represent all data collected at 20 Hz, while the larger red dots represent the last three data points at each dV level, which are used to calculate signal at the given dV level; and b) signal fraction remaining at each voltage difference (dV). Data points in light red, excluded from the sigmoidal fit, are signals from each dV level that do not meet quality control metrics.

**Comment 5: Line 327-328: The authors mentioned potential thermal decomposition in desorption and GC analysis, how about potential fragmentation in CIMS?**

Response: To the best of our understanding, we believe that the reviewer refers to mass spectral fragmentation or ion decomposition reactions instead of thermal fragmentation in CIMS. We did observe ionization chemistry beyond iodide-adduction formation such as molecular fragmentation in this study. The coupled TAG-CIMS in this work has the advantage of investigating ionization chemistry by the separation of analytes from a mixture. If a chromatographic peak of a compound is well-resolved in CIMS, all signals detected from the ionization of a single analyte are observed at the same chromatographic retention time and unambiguously assignable to that specific compound, including iodide adducts, products of adduct declustering, fragments (generally not from iodide clustering), and any ions produced by simultaneous alternate chemistry with other ions present in the atmospheric pressure interface (e.g., air). This provides a clean mass spectrum for each chromatographically well-resolved analyte and is particularly useful when analytes are in a complex mixture. From the results of the clean mass spectrums, we found some ions in the non-adduct region (i.e., the other side of the iodide valley) of the mass defect plot and published them in another work (Bi et al., 2021). However, we believe that more work is needed to more systematically investigate fragmentation patterns in CIMS.

Comment 6: Figure 3: I suggest the authors add more space between different species on X axis. They are too close from each other in the current version. It would be better to add a legend for the circles, black lines and the boxes for a more straight forward interpretation of the figure.

Response: We thank the reviewer's suggestions for improving the clarity of Figure 3. We have increased the width of the figure to add more space between chemical species on x axis. The additional legend for the circles and lines are also added to the figure.

---

## Author Comment (AC2)

The authors would like to thank the reviewers for the feedback on the manuscript. We have made revisions to the manuscript according to the reviewers' comments and the extra experimental findings. The colorings of text in the reviewer response are:

- Light blue: Original reviewer comments
- Dark blue: Text added in the revision while  are the text deleted in the revised manuscript.
- Black: Original text in the submitted version of the manuscript and authors' response to the comments and others.

Note that the line number in the response is based on the revised clean-version manuscript.

Reviewer 1:

The authors investigated the calibration method for the Iodide CIMS with a Thermal desorption aerosol gas chromatograph (TAG) and an FID detector, allowing the determination of isomer-resolved sensitivity. For the same formula, sensitivities for different isomers were found to vary by 1-2 orders of magnitude. The results suggested calibration based on direct air sampling can be biased towards isomers with higher sensitivity. Sensitivity estimation using voltage scanning method after GC separation was compared to direct calibration (without column separation) and showed a high uncertainty by 0.5-1 order of magnitude. They also found that iodide CIMS sensitivity correlates with GC retention time, however, more work is needed for a calibration purpose.

This paper is well written with informative description. Results of the study can be useful for future applications of similar method. I have a few specific comments.

Specific comments:

Comment 1: This study measured particle phase OH and ozone oxidation products. These products are likely heavier and more oxidized/functionalized (thus less volatile) than gas phase compounds. Can the authors comment on the applicability of these calibration techniques to gas phase measurements, especially CIMS is often used for gas phase measurement? Would they expect similar results? Are there any suggestions for future applications?

Response: We would like to thank the reviewer first for thinking this manuscript useful for future applications. We believe that the calibration techniques (both the FID calibration approach for CIMS and the voltage scanning approach) can be applied to gas-phase measurements with some limitations.

Calibrating CIMS using FID relies on the GC separation of the analytes since FID itself cannot resolve individual chemical species from a mixture. Therefore, as long as the analyte, no matter it is in the gas- or phase-phase, can be collected by the upstream GC, separated by the GC column, and transferred to both detectors, its CIMS sensitivity can be quantified by the technique. While the thermal desorption aerosol gas chromatograph (TAG) was used for sample collection and GC separation in this work, any GC-based instrument can be used to apply this technique and the design and configurations of the GC will depend on the specific applications of the user. In this particular GC, TAG, was configured with an impactor cell that collects mostly particle-phase samples in air. However, previous work has shown that a filter cell can be used, which allows simultaneous collections of particle- and gas-phase samples (Isaacman et al., 2014). Additionally, recently developments on coupled GC-proton transfer reaction (PTR)-MS have shown that coupling a CIMS to a GC can analyze gas-phase analytes with the resolution of individual molecules (Claflin et al., 2021). Although it is not an iodide CIMS, it shows the potential to analyze gas-phase components in similar cases.

To strengthen these suggestions for future applications, we have revised the manuscript on Line 319:

"Notably, although this instrument TAG-CIMS/FID is specifically configured to collect only particle-phase samples, this calibration technique for quantifying CIMS sensitivity could be applied to gas-phase samples using a different instrument configuration and should be applicable to any GC-based instrument employed upstream of the two detectors as long as the analyte can be collected by the upstream instrument, separated by the GC column, and transferred to both detectors."

As for the voltage scanning techniques, it can be applied to a CIMS with direct air sampling as demonstrated by Isaacman-Vanwertz et al. (2018) and Mattila et al. (2020). The work in this study is to quantify the uncertainty for this calibration technique. Therefore, this work will not limit its application to particle-phase only scenarios.

Comment 2: Can the authors add some discussion on the influence of RH on the different calibration methods? It is known that sensitivity is RH dependent for I-CIMS for many chemicals.

Response: We agree with the reviewer and have added a discussion of its influence in the context of a TAG-CIMS/FID. In brief, the relative humidity (RH) influence is minimal in coupling a GC to a CIMS, which is one merit of the coupled GC-CIMS but was not mentioned in the submitted version of the manuscript. Water vapor is not collected by the TAG impactor, and any residual moisture is purged before transfering analytes to the GC column. Analytes are transferred to the CIMS using ~1 sccm of carrier gas (i.e., helium), and this GC flow is what the CIMS samples, so the RH of the CIMS samples flow is always the same (~0%). RH consequently will not impact the CIMS sensitivities of analytes quantified by the FID calibration method. Since the TAG does not introduce moisture to the ion-molecular reaction region (IMR) of the CIMS, the RH in the IMR can be controlled at a fixed level by adding water vapor in the ~2 lpm reagent ion flow. A benefit of the proposed approach is thus that future work could be done to quantitatively probe sensitivity dependencies of RH using the coupled TAG-CIMS/FID by feeding the IMR with reagent ion flow containing different RH levels. To highlight this merit of TAG-CIMS/FID, we have revised the manuscript on Line 346:

"It is well-established that sensitivity is humidity-dependent for many chemicals in an iodide CIMS (Lee et al., 2014). Because analytes entering the IMR come from the GC in a dry helium flow, the relative humidity in the IMR is stable and can be controlled by adjusting the mixed water vapor in the reagent ion flow. Therefore, the coupled TAG-CIMS/FID provides opportunities for future work to quantitatively investigate the humidity dependency of sensitivity of those chemicals."

Comment 3: Line 142: "similar" should be "similarly".

Response: We have revised the manuscript on Line 142

"…is operated similarly to typical direct air sampling by CIMS…"

Comment 4: Figure 1: This figure is a little complicated to understand. The authors have very thorough method description on line 190-205. However, for audience that are not very familiar with the technique and the concept, it is overly technical, and they may get lost through the text. One suggestion for the authors is to make a plot showing the technique, especially the voltage scanning method coupled to GC separation, and the corresponding data collection and quantification that were used for constructing Figure 1. This can be a cartoon/plot illustration in the supporting information.

Response: We apologize for making a figure too hard to understand. We have also revised the discussions associated with Figure 1 to improve the clarity of the manuscript on Line 174:

"This timescale is not practical for GC applications, in which chromatographic peak widths are typically less than tens of seconds.  In this work,  instrument used in this work was able to switch and stabilize all voltages in ~100 ms. Using these conditions as an example, we develop a rough understanding of the maximum possible rate of voltage scanning. If voltages are varied at 5 Hz (i.e., 200 ms per voltage level), and data is collected at 10 Hz (i.e., 100 ms per data point), data would alternate between single points of "transition spectra" that must be ignored, and single points of real spectra representing the new voltage level. Such an approach is impractical, however, as each level would be represented by a single spectrum, collected immediately after the transition. Instead, to achieve a reasonable and accurate relationship between the signal fraction remaining and the dV, we switch voltages at a slower rate (2.5 Hz) and acquire data at a faster rate (20 Hz~~Generally, the voltages need to be switched fast enough to have multiple dV levels across one chromatographic peak, but slow enough to reach a steady state before switching again. Data acquisition must occur faster than voltage switching, with sufficient time resolution to discard any data collected during voltage transitions (i.e., non-steady-state). In this study, these requirements were met by switching voltages at 2.5 Hz and acquiring data at 20 Hz. With this acquisition rate, eight (i.e. 20 Hz/ 2.5 Hz) data points per dV were collected, with at least the first 2-3 as "transition spectra" that need to be ignored; practically speaking we find that only the last 3-4 spectra are stable (typical relative standard deviation < 20%), so the first 5 spectra of each level are ignored and the signal at a given dV level is taken as the average of the final 3 spectra collected. We note that voltage switching and data acquisition rates are likely instrument specific and optimal settings may vary significantly across different CIMS instruments.

The voltage settings of the CIMS in the regular mode are used as a set of baseline values (designated as dV=2 V). Fourteen different sets of voltage settings, each of which has a constant voltage deviation from the baseline values ( -0.5 V to +12 V). The voltage setting is varied at 2.5 Hz, alternating between the baseline values and a set of voltages representing a different dV level. The voltages upstream of the second quadrupole moved simultaneously with the change of dV to maintain a constant electric field gradient across the quadruples and consequently minimize impacts on ion transmission efficiency (Lopez-Hilfiker et al., 2016). As shown in the upper plot of Figure 1a, the set of dV levels is always in the same order, but is not monotonic, randomized to avoid the influence of potential memory effects on the results. An example of the output data for a signal chromatographic peak is shown in Figure 1a. Grey dots represent all data collected at 20 Hz, while the larger red dots represent the last three spectra at each dV level, which are used to calculate signal at that voltage scan with the given dV level."

As suggested by the review, we have also added a time-series of the voltage settings in Figure 1 to improve clarity of the method.

[Figure]

**Figure 1**. Demonstration of the voltage scan method to the GC-CIMS with a) recreation of chromatographic peak (grey dots represent all data collected at 20 Hz, while the larger red dots represent the last three data points at each dV level, which are used to calculate signal at the given dV level; and b) signal fraction remaining at each voltage difference (dV). Data points in light red, excluded from the sigmoidal fit, are signals from each dV level that do not meet quality control metrics.

**Comment 5:** Line 327-328: The authors mentioned potential thermal decomposition in desorption and GC analysis, how about potential fragmentation in CIMS?

Response: To the best of our understanding, we believe that the reviewer refers to mass spectral fragmentation or ion decomposition reactions instead of thermal fragmentation in CIMS. We did observe ionization chemistry beyond iodide-adduction formation such as molecular fragmentation in this study. The coupled TAG-CIMS in this work has the advantage of investigating ionization chemistry by the separation of analytes from a mixture. If a chromatographic peak of a compound is well-resolved in CIMS, all signals detected from the ionization of a single analyte are observed at the same chromatographic retention time and unambiguously assignable to that specific compound, including iodide adducts, products of adduct declustering, fragments (generally not from iodide clustering), and any ions produced by simultaneous alternate chemistry with other ions present in the atmospheric pressure interface (e.g., air). This provides a clean mass spectrum for each chromatographically well-resolved analyte and is particularly useful when analytes are in a complex mixture. From the results of the clean mass spectrums, we found some ions in the non-adduct region (i.e., the other side of the iodide valley) of the mass defect plot and published them in another work (Bi et al., 2021). However, we believe that more work is needed to more systematically investigate fragmentation patterns in CIMS.

**Comment 6:** Figure 3: I suggest the authors add more space between different species on X axis. They are too close from each other in the current version. It would be better to add a legend for the circles, black lines and the boxes for a more straight forward interpretation of the figure.

Response: We thank the reviewer's suggestions for improving the clarity of Figure 3. We have increased the width of the figure to add more space between chemical species on x axis. The additional legend for the circles and lines are also added to the figure.

[Figure]

**Figure 3.** Sensitivities of constituent isomers of formulas for which at least two isomers had sensitivities obtained in the oxidation experiments. Each circle shows the sensitivity of an isomer and the area of circle represents the mole fraction of the isomer in the formula. Box represents the first to the third quartile. Black lines are the median values of the sensitivities.

*: unit converted for direct-air-sampling CIMS using 100 mbar in IMR, 2 slpm sample flow rate, and 2 slpm reagent ion flow rate.

Comment 7: Line 356: It would be useful if the authors can provide a list of these products in a table in SI. Are they the same chemicals presented in Figure 3?

Response: We apologize for the lack of clarity in presenting the compounds for Figure 3 and 4. The compounds shown in Figure 3 and 4 have some overlaps but are not identical. Only formulas containing 2 or more isomers with obtained sensitivity are included in Figure 3 while compounds presented in Figure 4 need to have both obtained sensitivity and $dV_{50}$. As suggested by the reviewer, we have added a Table S1 and S2 to list compounds presented in Figure 3 and 4, respectively.

Table S1. The elemental formula and sensitivity of compounds presented in Figure 3. Note that only formulas containing 2 or more isomers with calculated sensitivity are included.

| Oxidation experiment | Elemental formula | Isomer No. | Sensitivity (ions/mole/million reagent ions) |
|---|---|---|---|
| Limonene-O₃ | C7H10O3 | 1 | 1.77E+16 |
| | | 2 | 4.83E+14 |
| | C8H12O3 | 1 | 1.34E+16 |
| | | 2 | 1.64E+14 |
| | C9H10O2 | 1 | 3.72E+14 |
| | | 2 | 1.38E+16 |
| | C9H12O4 | 1 | 4.37E+15 |
| | | 2 | 8.20E+15 |
| | | 3 | 2.43E+15 |
| | | 4 | 6.58E+15 |

| | | | |
|---|---|---|---|
| | | 5 | 1.26E+16 |
| | | 6 | 8.36E+16 |
| | C9H14O3 | 1 | 1.26E+16 |
| | | 2 | 7.86E+16 |
| | | 3 | 1.34E+16 |
| | | 4 | 7.18E+15 |
| | | 5 | 1.37E+17 |
| | C9H16O3 | 1 | 1.32E+16 |
| | | 2 | 1.33E+16 |
| | C10H14O3 | 1 | 1.23E+16 |
| | | 2 | 2.94E+16 |
| | | 3 | 1.79E+16 |
| | C10H16O4 | 1 | 9.11E+16 |
| | | 2 | 1.16E+17 |
| | | 3 | 2.27E+16 |
| | | | |
| Limonene-OH | C5H6O4 | 1 | 1.09E+15 |
| | | 2 | 2.20E+14 |
| | | 3 | 8.70E+14 |
| | C7H10O3 | 1 | 2.54E+16 |
| | | 2 | 1.21E+16 |
| | C7H10O4 | 1 | 4.03E+16 |
| | | 2 | 2.48E+16 |
| | C8H10O4 | 1 | 2.67E+16 |
| | | 2 | 5.44E+16 |
| | C8H8O4 | 1 | 4.91E+14 |
| | | 2 | 5.83E+15 |
| | C9H12O4 | 1 | 5.04E+15 |
| | | 2 | 3.05E+16 |
| | | 3 | 4.13E+16 |
| | | 4 | 4.90E+15 |
| | | 5 | 1.54E+16 |
| | C9H14O3 | 1 | 6.37E+15 |
| | | 2 | 9.28E+15 |
| | | 3 | 4.75E+15 |
| | C9H14O4 | 1 | 3.09E+16 |
| | | 2 | 2.56E+16 |
| | | 3 | 1.30E+16 |
| | | 4 | 6.46E+14 |
| | | 5 | 1.52E+15 |
| | C10H14O3 | 1 | 4.83E+15 |
| | | 2 | 8.15E+15 |
| | C10H16O4 | 1 | 8.13E+16 |
| | | 2 | 2.52E+16 |
| | | 3 | 6.63E+16 |
| | | 4 | 1.75E+16 |
| | | | |
| TMB-OH | C8H10O4 | 1 | 1.33E+15 |
| | | 2 | 6.03E+16 |

| | C8H12O4 | 1 | 9.67E+15 |
|---|---|---|---|
| | | 2 | 3.55E+16 |
| | C9H12O4 | 1 | 1.84E+15 |
| | | 2 | 8.78E+15 |
| | | 3 | 6.58E+16 |
| | C9H12O5 | 1 | 3.16E+15 |
| | | 2 | 1.15E+16 |
| | C9H14O4 | 1 | 6.83E+16 |
| | | 2 | 7.40E+16 |
| | C9H14O5 | 1 | 1.38E+16 |
| | | 2 | 1.06E+16 |
| | | 3 | 4.40E+15 |
| | | 4 | 2.00E+15 |
| | | 5 | 1.38E+15 |

Table S2. The elemental formula, sensitivity, and $dV_{50}$ of compounds presented in Figure 4. Note that only compounds with both calculated sensitivity and $dV_{50}$ are included.

| Oxidation experiment | Compound No. | Elemental formula | Sensitivity (ions/mole/million reagent ions) | | $dV_{50}$ (V) | |
|---|---|---|---|---|---|---|
| | | | Mean | Standard deviation | Mean | Standard deviation |
| Limonene-O$_3$ | 1 | C5H6O4 | 6.10E+14 | 8.67E+13 | 6.21 | 0.33 |
| | 2 | C7H10O2 | 1.23E+15 | 2.67E+14 | 5.49 | 0.23 |
| | 3 | C7H10O3 | 4.83E+14 | 1.16E+14 | 5.09 | 0.29 |
| | 4 | C7H8O4 | 1.46E+15 | 1.24E+14 | 6.44 | 0.35 |
| | 5 | C8H12O3 | 1.34E+16 | 4.52E+14 | 5.86 | 0.48 |
| | 6 | C8H12O3 | 1.64E+14 | 5.06E+13 | 5.37 | 0.25 |
| | 7 | C8H12O4 | 3.95E+16 | 7.77E+15 | 6.14 | 0.21 |
| | 8 | C9H10O2 | 3.72E+14 | 1.49E+14 | 5.19 | 0.30 |
| | 9 | C9H10O2 | 1.38E+16 | 1.83E+14 | 4.64 | 0.74 |
| | 10 | C9H12O4 | 4.37E+15 | 9.09E+14 | 4.93 | 0.36 |
| | 11 | C9H12O4 | 8.20E+15 | 4.30E+14 | 5.98 | 0.37 |
| | 12 | C9H12O4 | 2.43E+15 | 2.38E+14 | 4.71 | 0.74 |
| | 13 | C9H12O4 | 6.58E+15 | 1.98E+15 | 5.62 | 0.35 |
| | 14 | C9H12O4 | 1.26E+16 | 1.17E+15 | 4.53 | 0.47 |
| | 15 | C9H12O4 | 8.36E+16 | 3.14E+16 | 6.62 | 0.44 |
| | 16 | C9H14O3 | 1.26E+16 | 3.78E+15 | 4.87 | 0.60 |
| | 17 | C9H14O3 | 7.86E+16 | 5.34E+15 | 6.25 | 0.33 |
| | 18 | C9H14O3 | 1.34E+16 | 1.49E+15 | 5.19 | 0.48 |
| | 19 | C9H14O3 | 7.18E+15 | 1.50E+14 | 4.49 | 0.84 |
| | 20 | C9H14O4 | 5.77E+16 | 3.57E+15 | 6.39 | 0.62 |
| | 21 | C9H16O3 | 1.32E+16 | 2.58E+14 | 5.43 | 0.26 |
| | 22 | C9H16O3 | 1.33E+16 | 1.25E+15 | 5.20 | 0.74 |
| | 23 | C10H14O3 | 1.23E+16 | 9.40E+14 | 5.08 | 0.73 |
| | 24 | C10H14O3 | 2.94E+16 | 2.78E+15 | 5.81 | 0.32 |
| | 25 | C10H14O3 | 1.79E+16 | 1.51E+15 | 5.66 | 0.42 |
| | 26 | C10H16O4 | 9.11E+16 | 1.09E+16 | 6.96 | 0.26 |

| | | | | | | |
|---|---|---|---|---|---|---|
| | 27 | C10H16O4 | 1.16E+17 | 2.67E+16 | 7.51 | 0.22 |
| | 28 | C10H16O4 | 2.27E+16 | 5.54E+15 | 5.97 | 0.32 |
| | | | | | | |
| Limonene-OH | 1 | C5H6O4 | 1.09E+15 | 1.01E+14 | 4.97 | 0.41 |
| | 2 | C5H6O4 | 8.7E+14 | 4.30E+14 | 3.61 | 0.18 |
| | 3 | C7H10O3 | 2.54E+16 | 1.24E+16 | 4.74 | 0.52 |
| | 4 | C7H10O3 | 1.21E+16 | 8.53E+14 | 5.64 | 0.24 |
| | 5 | C7H10O4 | 4.03E+16 | 4.31E+15 | 5.57 | 0.24 |
| | 6 | C7H10O4 | 2.48E+16 | 4.69E+15 | 5.64 | 0.22 |
| | 7 | C8H10O4 | 2.67E+16 | 3.69E+15 | 4.41 | 0.41 |
| | 8 | C8H8O4 | 5.83E+15 | 1.15E+15 | 3.72 | 0.61 |
| | 9 | C9H12O4 | 5.04E+15 | 9.40E+14 | 5.27 | 0.25 |
| | 10 | C9H12O4 | 3.05E+16 | 5.46E+15 | 5.34 | 0.33 |
| | 11 | C9H12O4 | 4.13E+16 | 8.40E+15 | 5.66 | 0.53 |
| | 12 | C9H14O3 | 6.37E+15 | 1.99E+13 | 5.35 | 0.94 |
| | 13 | C9H14O4 | 3.09E+16 | 3.65E+15 | 5.38 | 0.23 |
| | 14 | C9H14O4 | 2.56E+16 | 8.96E+15 | 5.73 | 0.43 |
| | 15 | C9H14O4 | 1.3E+16 | 1.25E+15 | 4.89 | 0.46 |
| | 16 | C9H14O4 | 1.52E+15 | 2.21E+14 | 5.44 | 0.86 |
| | 17 | C10H16O4 | 8.13E+16 | 9.26E+15 | 6.56 | 0.24 |
| | 18 | C10H16O4 | 2.52E+16 | 3.99E+15 | 5.06 | 0.35 |
| | 19 | C10H16O4 | 6.63E+16 | 6.88E+15 | 6.39 | 0.43 |
| | 20 | C10H16O4 | 1.75E+16 | 1.28E+15 | 5.35 | 0.50 |
| | | | | | | |
| TMB-OH | 1 | C8H10O4 | 1.33E+15 | 1.23E+14 | 3.59 | 0.67 |
| | 2 | C8H10O4 | 6.03E+16 | 2.05E+15 | 5.72 | 0.47 |
| | 3 | C8H12O4 | 9.67E+15 | 3.38E+15 | 4.11 | 0.60 |
| | 4 | C8H12O4 | 3.55E+16 | 5.41E+15 | 4.35 | 0.58 |
| | 5 | C9H12O4 | 1.84E+15 | 9.02E+14 | 4.84 | 0.20 |
| | 6 | C9H12O4 | 8.78E+15 | 8.04E+14 | 4.55 | 0.45 |
| | 7 | C9H12O4 | 6.58E+16 | 3.11E+16 | 5.41 | 0.37 |
| | 8 | C9H12O5 | 3.16E+15 | 1.37E+15 | 6.51 | 0.38 |
| | 9 | C9H12O5 | 1.15E+16 | 5.16E+15 | 5.54 | 0.22 |
| | 10 | C9H14O4 | 6.83E+16 | 2.98E+16 | 5.38 | 0.46 |
| | 11 | C9H14O4 | 7.40E+16 | 3.06E+16 | 6.73 | 0.24 |
| | 12 | C9H14O5 | 1.38E+16 | 3.50E+15 | 6.73 | 0.31 |
| | 13 | C9H14O5 | 1.06E+16 | 4.87E+15 | 6.31 | 0.25 |
| | 14 | C9H14O5 | 4.40E+15 | 1.09E+15 | 5.67 | 0.44 |
| | 15 | C9H14O5 | 1.38E+15 | 2.25E+14 | 5.52 | 0.99 |

Comment 8: Line 358: What is the influence of OH level on the formed products or instrument sensitivity? Are there more oxygenated species or fragmentation at high OH? Please add some discussion.

Response: The reviewer raised two questions in this comment: what is the influence of OH level on 1) instrument sensitivity of the TAG-CIMS and 2) formed oxidation products.

For the first question, instrument sensitivity is a function of the physical-chemical properties of the molecules (e.g., functional groups, volatility, and polarity) and instrumental setups (e.g., RH, pressure,

and voltage settings). There is no direct impact of OH level on the instrument (i.e., the CIMS is not exposed to the OH level, just to the products), so it is not clear there would be any impact on instrument response. The instrument sensitivity should always be a constant for a specific molecule under a fixed instrumental condition. In fact, the direct measured CIMS sensitivity in this study is the average of isomer sensitivity at two or more OH levels for limonene-OH and TMB-OH experiments. As shown by the error bars of the isomer sensitivity in Figure 4, the uncertainties of instrument sensitivity across three OH levels are mostly within 30% and are mostly the result of measurement uncertainties.

However, it is known that the higher OH level will likely generate more oxidized compounds in an oxidation flow reactor (OFR) (Lambe et al., 2012). Although it is out of the scope of the current study, we also observed the increase of O/C ratio with the increase of OH levels. More importantly, we found a change of isomer composition of a formula with the increase of OH level. As shown in Figure 3, isomers in a formula may have significantly different instrument sensitivity. The change of isomer composition in a formula may impact the average sensitivity of a formula, which is typically observed in a CIMS with direct air sampling. In this case, the average sensitivity of the formula will likely increase at higher OH levels. Since the focus of this work is not on the influence of OH levels on isomer composition, we have revised the manuscript to briefly discuss the potential future application of this instrument on investigating OH-dependent isomer composition, but not elaborate on the details of the findings. Line 323 has been revised:

"Additionally, the coupled TAG-CIMS/FID can be applied to investigate the change of isomer composition with the increase of OH levels in future studies. It is possible that the isomers of a formula produced at higher OH levels are more oxidized compounds thus changing the isomer distributions in the formula. In this case, the average sensitivity of the formula will likely increase at higher OH levels."

Comment 9: Figure 8 y axis: missing space after 'log'

Response: We would like to thank the reviewer for pointing out the mistake. We have revised Figure 8 as shown below.

[Figure]

**Figure 8.** The relationship between differences (Δ) in log(sensitivity) and retention index for all pairs of isomers with a given formula. Black markers are all data equally divided into eight bins (octiles), centered on averages with error bars representing standard deviations.

Comment 10: Figure S1: Why is the baseline higher for OH level 1 in all four panels?

Response: We thank the reviewer for pointing out the differences between baselines in Figure S1. However, we are not totally sure about the reason for the higher baseline. Here, we provide a possible explanation for the observation. We think that the elevated baseline in Figure S1 is due to the differences in the total mass of sampled analytes across different OH levels. The increase of sample amount can raise 'baseline' since the 'baseline' observed in a GC chromatogram is the unresolved signal of many smaller peaks and the increase in the total mass of analytes may enhance the stacking effects of small background peaks. Although the detailed chemical analysis of the analytes across different OH levels is out of the scope of this study, total mass of analytes collected in the sampling cell is likely not the same at each OH level. Therefore, some differences in the baseline could happen.

Reviewer 2:

Summary:
Organic aerosol reaction products from limonene+O3 and trimethylbenzene+OH were measured using a custom thermal-desorption GC instrument with the GC eluant split between an FID and a chemical ionization mass spectrometer using I- CI. The purpose was to understand the accuracy of the so-called "voltage scanning" method of I- CIMS calibration, in which collision-induced dissociation is used to quantify the ion-molecule adduct binding energy, which supposedly is linearly correlated with sensitivity. The assessment was done by comparing the voltage scan parameter dV50 to sensitivity determined by a combination of FID and chemical formula (from CIMS). The findings are that sensitivities vary greatly between isomers; dv50 is a poor predictor of sensitivity for individual compounds, but may have some use when many compounds are summed; and GC retention time has limited relationship with sensitivity.

Major comments:
The manuscript is well-written and well-organized with a nice attention to detail. The subject matter is certainly worthy of publication, since it addresses a useful and somewhat controversial question in the CIMS community. Many of my questions on a first reading were answered later in the manuscript.

I have one major issue with this manuscript, and several overall questions:
Comment 1: Major issue: From Figure 4, it really does not look like the voltage scanning method works especially well. It looks slightly better if some averaging is done (Figure 5, although I have some reservations about the mathematical appropriateness of the average -see below). However, it is hardly useful for individual compounds. This is surprising since the theory seems solid, and previous work indicated that there was certainly a relationship between the voltage scan parameter and sensitivity. What is missing from this manuscript is some discussion of why the experimental relationship is so weak. The experiment was designed well, but are there some experimental uncertainties that could explain the results? Or is the theory "wrong" – meaning that dv50 does not correlate with binding energy, and/or binding energy does not correlate with sensitivity for some compounds? If so, why? Even some speculation in the discussions section would be useful, since it could provide some ideas for how to improve the method.

Response: We appreciate the reviewer for carefully reading the manuscript and thinking this subject matter is worthy of publication. Many comments provided by the reviewer are very insightful suggestions to improve the quality of the manuscript and may potentially broaden the understanding of the voltage scanning method for the entire CIMS community. However, we notice that some questions brought up in the comments are related to the original theoretical development (Iyer et al., 2016) and experimental test of voltage scanning (Lopez-Hilfiker et al., 2016), which were not developed by the authors or under this PI. In this work, we tried implementing this voltage scanning method using similar instrumental setting as the published work and compared the estimated sensitivity with directly measured ones to validate the approach for a broader range of chemical species. Below we address the reviewer's concerns for those questions within the context of this study.

We agree with the reviewer that the voltage scanning method has limited utility for individual compounds given the large uncertainty found in this study. We have drawn similar conclusions as stated on Line 435: "The results show that about 60% and 80% of compounds are estimated within a factor of 3 and 10 uncertainties, respectively, indicating that the voltage scan approach has high uncertainties for individual components."

We believe that there are three possible reasons for the mismatch between theory and experiment: 1) the empirical nature of the log-linear relationship, 2) the original uncertainty in the empirical relationship, and 3) the application of the techniques to a wider range of chemical species. We note that the log-linear correlation between CIMS sensitivity and the calculated binding enthalpy is empirical. The theory works under two important premises: (1) sensitivity is log-linearly correlated with the enthalpy of the iodide-adduct bond and (2) the enthalpy of the bond can be represented by the $dV_{50}$ in the voltage scans. As excerpted from Iyer et al. (2016), "the proposed fit to the binding enthalpy is thus purely empirical, with no deeper physical significance beyond the simple fact that the survival probability of an ion−molecule cluster must increase with the strength of its binding." The first premise is purely empirical and it is possible that the log-linear relationship represents an upper limit of the possible sensitivity of analytes. Some additional unknown factors might have influences on sensitivity beyond the bonding enthalpy thus decreasing the sensitivity of some analytes from the upper limit (e.g., analytes below the log-linear line in Figure 4).

Even if we assume the relationship holds, large uncertainty exists in the empirical log-linear fit between sensitivity and the binding enthalpy. In their work, they observed 3 compounds (i.e., oxalic acid,

methylerythritol, and methacrylic acid) that did not match the log-linear fit, out of the 13 analytes (23% mismatch). Their results show that 77% (versus 60% in this work) and 85% (versus 80% in this work) of compounds are estimated within a factor of 3 and 10 uncertainties, respectively. Therefore, large uncertainties exist in the original relationship and that uncertainty is in rough agreement with the larger set of compounds tested here.

Additionally, the experimental validation of the theory which transformed the binding enthalpy into $dV_{50}$ was conducted for a limited number of liquid standards, which were mostly mono- and di-acids. It was unknown whether this relationship can hold for a wider range of oxidation products and this motivated us to study the application of the voltage scanning method to those chemical species in this study. As found in this work, the uncertainty is high for individual compounds, but the voltage scanning method can still provide reasonably accurate results when total concentrations of all compounds or a class of compounds are the primary interest. We have revised the manuscript on Line 439 to reflect our thoughts on the mismatch between the previous and current study.

"The results show that about 60% and 80% of compounds are estimated within a factor of 3 and 10 uncertainties, respectively, indicating that the voltage scan approach has high uncertainties for individual components. The uncertainty for individual compounds found in this work is similar to the studies originally proposing this log-linear relationship (i.e., 77% and 85% of compounds are estimated within a factor of 3 and 10 uncertainties, respectively) (Iyer et al., 2016). The uncertainty is likely caused by the transfer of uncertainty from the empirical correlation itself and the consideration of wider range of chemical species. It is also possible that optimizing certain instrumental parameters could improve the accuracy of the voltage scanning method, but future work is needed to investigate the optimization method further."

 Overall questions:

Comment 2: 1.The voltage difference between the skimmer and the BSQ affects not only the molecule-adduct declustering environment, but also the transmission efficiency of ions into the BSQ. Do you have a sense of the relative magnitude of these effects? Could it be important to normalize to some metric of transmission efficiency?

Response: We did observe the change of total reagent ions ($I^-$ + $IH2O^-$) with different voltage settings, which likely reflect some influences on the transmission efficiency by the voltage scanning.  We were aware of some debates in the field about whether normalization to reagent ions is necessary, but chose to follow the original approach reported in the literature. Since the voltage scanning method was previously developed by Lopez-Hilfiker et al. (2016), we follow similar methods as used in their study. It is known that the mass transmission function is dependent on the axial voltage gradient along the quadrupole rods (Heinritzi et al., 2016). To maintain a constant transmission efficiency, the electric potentials upstream of the BSQ moved simultaneously with the change of dV so that the declustering strength was incrementally changed while keeping a constant gradient across the quadrupoles. We have revised the manuscript on Line 188 to note the potential impacts on transmission efficiency:

"Fourteen different sets of voltage settings, each of which has a constant voltage deviation from the baseline values (-0.5 V to +12 V). The voltage setting is varied at 2.5 Hz, alternating between the baseline values and a set of voltages representing a different dV level. The voltages upstream of the second quadrupole moved simultaneously with the change of dV to maintain a constant electric field gradient across the quadruples and consequently minimize impacts on ion transmission efficiency (Lopez-Hilfiker et al., 2016)."

Comment 3: Additionally, in lines 143-145, it seems you are normalizing to the reagent ion signal, but this also changes during the voltage scan. Could this distort the shapes of the sigmoid curves, since you

are not measuring directly the change in signal, but rather the change in signal relative to the change in I-signal (or sum of I- and IH2O-, not sure which was used here)?

Response: We apologize for lack of clarity suggesting that the analyte signals were normalized to reagent ion signal during voltage scans. In fact, the signals were normalized to the reagent ion only in regular mode and not in voltage scan mode. As suggested by the reviewer, the reagent ion signal in regular mode was an almost constant value but can vary significantly during the voltage scan. We agree with the reviewer that normalizing the analyte signals to the reagent ion will distort the shapes of the sigmoid curves. Since we did not actually do that, we have revised the manuscript to avoid such misunderstandings on Line 247:

"Integration of CIMS peaks yields units of CIMS-response $\times$ s, where CIMS response is ions/s (i.e., counts per second, "cps") normalized to the number of reagent ions (typically in millions). CIMS peak areas are therefore in the units of (ions/million reagent ions) /s $\times$ s = ions/million reagent ions. Normalization to the reagent ion was not used for generating voltage scanning curves, following the published approach (Lopez-Hilfiker et al., 2016)"

Comment 4: Second, I have some related questions about the accuracy and precision of this method. a) How sensitive is this method to instrumental noise? Or, in other words, what signal-to-noise ratio in the actual measurement is required to achieve meaningful results? b) How sensitive is this method to the quality of the sigmoid fit? What is the typical uncertainty in the fit parameters, and how does this translate to uncertainty in the calculated sensitivity? c) How sensitive is this method to instrument settings? The chosen setting of V0 – the baseline skimmer-BSQ voltage – is probably quite important. Are there other instrument settings (pressures, voltages) that need to be exactly right for this method to work?

Response: The reviewer raises three questions about the accuracy and precision of this method, which we address in order below.

Firstly, we agree with the reviewer that an adequate signal-to-noise ratio is needed to achieve meaningful results in this study. The lowest signal-to-noise ratio of the compound that has a reported $dV_{50}$ in this study (i.e., smallest chromatographic peak height divided by the standard deviation of baseline signal) are about 30. This suggest that voltage scanning can be achieved at concentrations near, but not quite at, typical levels of quantitation (often considered as 6-10 times baseline noise). However, there are other factors such as the width of the peak and the timescale of the voltage scan can determine whether a reasonable sigmoid fit can be obtained. While the detailed response to the reviewer's comment on voltage scan timescale is in response to Comment 11, briefly speaking, the voltage scan needs to be fast enough to capture the peak yet cannot be too fast to lose stabilized signals under a given dV. Therefore, the accuracy and precision can be improved if the chromagraphic peaks are made wider by slowing down the rate of the GC temperature ramp, provided the widening of the peak does not lower the peak height to below some reasonable signal-to-noise.

Secondly, the uncertainty of $dV_{50}$ in the sigmoidal fit is 0.43, 0.55, and 0.47 in limonene-$O_3$, limonene-OH, and TMB-OH experiments, respectively. To compare the uncertainty in the fit with those in the measurements, we have added Table S2 to the supporting information, summarizing the uncertainty in the measured sensitivity, obtained from triplicated experiments, and $dV_{50}$, obtained from duplicated experiments. As shown in Table S2, the average standard deviations of measured $dV_{50}$ are 0.43, 0.42, and 0.44, in limonene-$O_3$, limonene-OH, and TMB-OH experiments, respectively, which roughly agree with

the uncertainty in the sigmoidal fit. The transfer of uncertainty from measured $dV_{50}$ to estimated sensitivity will be determined based on the slope of the log-linear fit and the region of the relationship where maximum sensitivity is reached. As fitted in Figure 5, log Sensitivity = $0.6 \times dV_{50} + 12.9$ and the sensitivity reach maximum at $dV_{50} = 6.9$ V. If $dV_{50}$ is greater than 6.9 V where maximum sensitivity is reached, the uncertainty in $dV_{50}$ will not impact the uncertainty in sensitivity. Conversely, the average 0.4 V standard deviation observed in this study corresponds to 0.24 log unit deviation in sensitivity, which is similar to the measurement error shown in Figure 4.

Finally, although we did not develop the voltage scanning method, we agree with the reviewer that it is possible that parameters such as baseline voltages and IMR pressures that are quite important to make the voltage scanning method work. As we discussed on Line 172: "No consensus currently exists on the rate at which voltages can (or should) be scanned, the number of spectra collected at each dV level, or the number or range of dV levels scanned". Similarly, no consensus exists on the operating conditions of the IMR, though there is some movement toward more standardized operation. We agree with the reviewer that the limitations of voltage scanning are still poorly understood by the community. We hope that the method we introduce in this work will in part enable a more thorough examination of the approach.

Table S2. The elemental formula, sensitivity, and $dV_{50}$ of compounds presented in Figure 4. Note that only compounds with both calculated sensitivity and $dV_{50}$ are included.

| Oxidation experiment | No. | Elemental formula | Sensitivity (ions/mole/million reagent ions) | | $dV_{50}$ (V) | |
| --- | --- | --- | --- | --- | --- | --- |
| | | | Mean | Standard deviation | Mean | Standard deviation |
| Limonene-$O_3$ | 1 | C5H6IO4 | 6.10E+14 | 8.67E+13 | 6.21 | 0.33 |
| | 2 | C7H10IO2 | 1.23E+15 | 2.67E+14 | 5.49 | 0.23 |
| | 3 | C7H10IO3 | 4.83E+14 | 1.16E+14 | 5.09 | 0.29 |
| | 4 | C7H8IO4 | 1.46E+15 | 1.24E+14 | 6.44 | 0.35 |
| | 5 | C8H12IO3 | 1.34E+16 | 4.52E+14 | 5.86 | 0.48 |
| | 6 | C8H12IO3 | 1.64E+14 | 5.06E+13 | 5.37 | 0.25 |
| | 7 | C8H12IO4 | 3.95E+16 | 7.77E+15 | 6.14 | 0.21 |
| | 8 | C9H10IO2 | 3.72E+14 | 1.49E+14 | 5.19 | 0.30 |
| | 9 | C9H10IO2 | 1.38E+16 | 1.83E+14 | 4.64 | 0.74 |
| | 10 | C9H12IO4 | 4.37E+15 | 9.09E+14 | 4.93 | 0.36 |
| | 11 | C9H12IO4 | 8.20E+15 | 4.30E+14 | 5.98 | 0.37 |
| | 12 | C9H12IO4 | 2.43E+15 | 2.38E+14 | 4.71 | 0.74 |
| | 13 | C9H12IO4 | 6.58E+15 | 1.98E+15 | 5.62 | 0.35 |
| | 14 | C9H12IO4 | 1.26E+16 | 1.17E+15 | 4.53 | 0.47 |
| | 15 | C9H12IO4 | 8.36E+16 | 3.14E+16 | 6.62 | 0.44 |
| | 16 | C9H14IO3 | 1.26E+16 | 3.78E+15 | 4.87 | 0.60 |
| | 17 | C9H14IO3 | 7.86E+16 | 5.34E+15 | 6.25 | 0.33 |
| | 18 | C9H14IO3 | 1.34E+16 | 1.49E+15 | 5.19 | 0.48 |
| | 19 | C9H14IO3 | 7.18E+15 | 1.50E+14 | 4.49 | 0.84 |
| | 20 | C9H14IO4 | 5.77E+16 | 3.57E+15 | 6.39 | 0.62 |
| | 21 | C9H16IO3 | 1.32E+16 | 2.58E+14 | 5.43 | 0.26 |
| | 22 | C9H16IO3 | 1.33E+16 | 1.25E+15 | 5.20 | 0.74 |
| | 23 | C10H14IO3 | 1.23E+16 | 9.40E+14 | 5.08 | 0.73 |
| | 24 | C10H14IO3 | 2.94E+16 | 2.78E+15 | 5.81 | 0.32 |
| | 25 | C10H14IO3 | 1.79E+16 | 1.51E+15 | 5.66 | 0.42 |
| | 26 | C10H16IO4 | 9.11E+16 | 1.09E+16 | 6.96 | 0.26 |

| | | | | | | |
|---|---|---|---|---|---|---|
| | 27 | C10H16IO4 | 1.16E+17 | 2.67E+16 | 7.51 | 0.22 |
| | 28 | C10H16IO4 | 2.27E+16 | 5.54E+15 | 5.97 | 0.32 |
| | | | | | | |
| | 1 | C5H6IO4 | 1.09E+15 | 1.01E+14 | 4.97 | 0.41 |
| | 2 | C5H6IO4 | 8.7E+14 | 4.30E+14 | 3.61 | 0.18 |
| | 3 | C7H10IO3 | 2.54E+16 | 1.24E+16 | 4.74 | 0.52 |
| | 4 | C7H10IO3 | 1.21E+16 | 8.53E+14 | 5.64 | 0.24 |
| | 5 | C7H10IO4 | 4.03E+16 | 4.31E+15 | 5.57 | 0.24 |
| | 6 | C7H10IO4 | 2.48E+16 | 4.69E+15 | 5.64 | 0.22 |
| | 7 | C8H10IO4 | 2.67E+16 | 3.69E+15 | 4.41 | 0.41 |
| | 8 | C8H8IO4 | 5.83E+15 | 1.15E+15 | 3.72 | 0.61 |
| | 9 | C9H12IO4 | 5.04E+15 | 9.40E+14 | 5.27 | 0.25 |
| Limonene-OH | 10 | C9H12IO4 | 3.05E+16 | 5.46E+15 | 5.34 | 0.33 |
| | 11 | C9H12IO4 | 4.13E+16 | 8.40E+15 | 5.66 | 0.53 |
| | 12 | C9H14IO3 | 6.37E+15 | 1.99E+13 | 5.35 | 0.94 |
| | 13 | C9H14IO4 | 3.09E+16 | 3.65E+15 | 5.38 | 0.23 |
| | 14 | C9H14IO4 | 2.56E+16 | 8.96E+15 | 5.73 | 0.43 |
| | 15 | C9H14IO4 | 1.3E+16 | 1.25E+15 | 4.89 | 0.46 |
| | 16 | C9H14IO4 | 1.52E+15 | 2.21E+14 | 5.44 | 0.86 |
| | 17 | C10H16IO4 | 8.13E+16 | 9.26E+15 | 6.56 | 0.24 |
| | 18 | C10H16IO4 | 2.52E+16 | 3.99E+15 | 5.06 | 0.35 |
| | 19 | C10H16IO4 | 6.63E+16 | 6.88E+15 | 6.39 | 0.43 |
| | 20 | C10H16IO4 | 1.75E+16 | 1.28E+15 | 5.35 | 0.50 |
| | | | | | | |
| | 1 | C8H10IO4 | 1.33E+15 | 1.23E+14 | 3.59 | 0.67 |
| | 2 | C8H10IO4 | 6.03E+16 | 2.05E+15 | 5.72 | 0.47 |
| | 3 | C8H12IO4 | 9.67E+15 | 3.38E+15 | 4.11 | 0.60 |
| | 4 | C8H12IO4 | 3.55E+16 | 5.41E+15 | 4.35 | 0.58 |
| | 5 | C9H12IO4 | 1.84E+15 | 9.02E+14 | 4.84 | 0.20 |
| | 6 | C9H12IO4 | 8.78E+15 | 8.04E+14 | 4.55 | 0.45 |
| | 7 | C9H12IO4 | 6.58E+16 | 3.11E+16 | 5.41 | 0.37 |
| TMB-OH | 8 | C9H12IO5 | 3.16E+15 | 1.37E+15 | 6.51 | 0.38 |
| | 9 | C9H12IO5 | 1.15E+16 | 5.16E+15 | 5.54 | 0.22 |
| | 10 | C9H14IO4 | 6.83E+16 | 2.98E+16 | 5.38 | 0.46 |
| | 11 | C9H14IO4 | 7.40E+16 | 3.06E+16 | 6.73 | 0.24 |
| | 12 | C9H14IO5 | 1.38E+16 | 3.50E+15 | 6.73 | 0.31 |
| | 13 | C9H14IO5 | 1.06E+16 | 4.87E+15 | 6.31 | 0.25 |
| | 14 | C9H14IO5 | 4.40E+15 | 1.09E+15 | 5.67 | 0.44 |
| | 15 | C9H14IO5 | 1.38E+15 | 2.25E+14 | 5.52 | 0.99 |

Comment 5: From figure 2 (and text) it is clear that many isomers may exist in chamber experiments (and also in ambient samples). What happens to the voltage scanning results when two or more isomers, with differing sensitivities, are present? Is this discernable in the shape of the sigmoid curve? At line 390 you state that the functional dV50 of an isomer mixture is equal to a signal-weighted average of the dV50 of each individual compound, but looking at the formula for a sigmoid function, it's not obvious to me how this is true. Can you show that the experimentally-fit dv50 of a mixture is equal to the weighted average of the individual species, especially when the baseline signal is different for each compound and unknown for a mixture?

Response: We thank the reviewer for making this comment which allows us to examine our averaging approach further. When a formula has a mixture of multiple isomers with varying signals, the analytical solution of the sigmoid fit of the mixture is complex and it is indeed not intuitive to examine, as suggested by the review. Rather, to examine the effect of combining sigmoidal curves, we use a simulated data approach described below. We find that the signal-weighted average of the $dV_{50}$ of each isomer is a reasonable approximation of the true $dV_{50}$ of two summed curves. A detailed description of the accuracy of the averaging technique, shown below, is added to the supporting information.

"**Calculation of $dV_{50}$ of a mixture of isomers**

When a formula has a mixture of multiple isomers with varying signals, the true $dV_{50}$ of the formula should be obtained using the sigmoid fit of the summed signal fraction remaining versus dV. However, to simplify the calculation of formula-based $dV_{50}$, this study applied signal-weighted average of $dV_{50}$ of each isomers to obtain the $dV_{50}$ of a formula. Here, we demonstrate this signal-weighted $dV_{50}$ is a good approximation of the true $dV_{50}$ using simulated data.

[Figure]

Figure S2. Simulated sigmoid fits of two isomers.

In Figure S2, we examine simulated sigmoid voltage scanning curves of two isomers within a formula ("red" and "blue") described by representative randomly selected coefficients. The signal fraction remaining ($SFR_{formula}^{dV_i}$) of the formula (i.e., the sum of the two isomers) at a given voltage setting ($dVi$ would described by the signal weighted average of the two curves:

$$SFR_{formula}^{dV_i} = \frac{S_{red}^{base}\left(SFR_{red}^{dV_i}\right)+S_{blue}^{base}\left(SFR_{blue}^{dV_i}\right)}{\left(S_{red}^{base}+S_{blue}^{base}\right)} \tag{15}$$

Where $S_{red}^{base}$ and $S_{blue}^{base}$ are signal of the two isomers at baseline voltages; $SFR_X^{dV_i}$ is the signal fraction remaining of each isomer at a given voltage setting.

Using Eq. 15, the expected sigmoidal curve can be obtained describing signal fraction remaining of the combined isomers for a given ratio of isomers. In other words, a signal fraction remaining curve can be generated for the formula by summing the two isomers at their given ratio. Rather than solving for $dV_{50}$ analytically (which may get complex for multiple isomers), the combined curve can be fit with a sigmoidal function to calculate the "true $dV_{50}$" that would be observed for the formula. This can be compared to the "signal-weighted $dV_{50}$" calculated as the signal-weighted average of the dV50 of the two isomers. Numerical solutions for two isomers are presented here to determine the accuracy of using a simplified signal-weighted $dV_{50}$ approach, across two orders of magnitude of relative ratio.

[Figure]

Figure S3. The distribution of a) true $dV_{50}$ and b) signal-weighted $dV_{50}$ of the formula with varying isomer abundance.

[Figure]

Figure S4. Comparison of true $dV_{50}$ and signal-weighted $dV_{50}$. The curve colored with red shows signals dominated by isomer Red while the blue section suggest that signals are dominated by isomer Blue. Color scale is logarithmic.

An arbitrary signal from 0-100 is assigned each to the "red" and "blue" isomers, implying two orders of magnitude differences in signals of the two isomers at baseline voltage. From Figure S3 it is clear that the

signal-weighted $dV_{50}$ is similar to the true $dV_{50}$ (calculated from sigmoid fit of Eqn 15) under the two orders of magnitude variance in isomer abundance. The difference between true and signal-weighted $dV_{50}$ (i.e., Figure S3a vs. S3b) is shown in Figure S4 as a function of the relative ratio of the isomers. When the signals of the two isomers are roughly equal (white region in the colored curved), the deviation reaches the maximum, but is still well within 10% and an absolute value of <0.5 V. On average, the deviation is only a few percent. This deviation is lower than the threshold of maximum relative standard deviation of $dV_{50}$ in duplicated samples, and is generally within the uncertainty of most fits, so is unlikely to contribute substantial uncertainty.

[Figure]

Figure S5. The average (a) absolute and (b) relative deviation of the true true $dV_{50}$ from the signal-weighted $dV_{50}$ for formulas with 2-20 isomers.

We generalize this result by numerically expanding to formulas with more than two isomers, generating a set of a given number of sigmoidal curve, each with randomly-assigned $dV_{50}$, sigmoidal rate coefficients, and relative signal between 3 – 9, 0.2 - 1, and 1 - 100, respectively. A Monte-Carlo analysis of 1000 such sets was conducted for each number of curves to examine the average absolute and relative deviation of the true $dV_{50}$ from the signal-weighted $dV_{50}$. The results suggest that the average absolute deviations are within 0.2 V and the average relative deviations are within 3% between the true and signal-weighted $dV_{50}$. No clear trend is observed in error with the increase in the number of curves, though the significant noise in the data may obscure any such trend. Therefore, we can conclude that the signal-weighted approach to calculate $dV_{50}$ of a formula is a good approximation of the true $dV_{50}$."

We have also made revisions on Line 410:

"The sensitivity of a formula is calculated as the average of isomer sensitivities weighted by their number of moles, while the $dV_{50}$ of a formula is calculated as the average of $dV_{50}$ weighted by their CIMS abundance (i.e., chromatographic peak area in CIMS data). Note that the signal-weighted method to calculate formula-based $dV_{50}$ is found to yield reasonable approximations of the true $dV_{50}$ and the detailed discussions are in the supporting information."

Comment 6: Is it even possible to use this quantification method when air is sampled directly, without pre-separation?

Response: To the best of our understanding, we believe that the quantification method referred to by the reviewer is the direct measurement of CIMS sensitivity using FID signals. In this case, we believe that it

is not possible to use the quantification method when air is sampled directly, without pre-separation of the GC. Because FID is a single-channel detector with responses nearly proportionally to the mass of carbons in the samples, the FID signals, without any pre-separation, are not molecule-resolved and is only a vague indicator of the total mass of carbon at a given time with corrections based on the average oxygen to carbon ratio. The information is too limited to generate reasonable quantification results but may be used for qualitative estimation of sensitivity in the future.

However, it is possible to temporarily couple the CIMS with a GC and an FID to measure individual sensitivities in a lab setting in cases where other calibration methods are not suitable. Once the sensitivities of the target analytes are determined, quantification can be achieved without the coupling of the GC and FID. As we have discussed on Line 332: "Once the CIMS sensitivity of a compound is obtained, quantification can be achieved for those compounds in other poor-signal conditions or even without the coupling of the FID."

Comment 7: And, from Figure 2, it seems the chromatographic peaks were not well-resolved, even for a single CI formula. If the chromatographic peaks overlap heavily, sensitivities vary greatly between isomers, and you are only able to accomplish one voltage scan per chromatographic peak, is this not problematic? How did you address this issue?

Response: We agree with the reviewer that it is difficult to perform this analysis on every peak due to the complex nature of the sample. Not all peaks are quantifiable and the use of voltage scan further increases the challenges for the analysis. Therefore, we focus on better-resolved peaks to compare sensitivity between direct FID measurement and voltage scans. Of the 82, 142, 158 peaks identified in CIMS in limonene-$O_3$, limonene-OH, and TMB-OH experiments, the $dV_{50}$ of 73, 115, and 110 peaks, respectively, can be obtained in duplicated experiments by the voltage scanning method. Therefore, it is indeed true that not all identified peaks have large enough peak width to obtain reasonable $dV_{50}$, but most of the peaks in this study have a measured $dV_{50}$. In cases where users are particularly interested in a set of compounds, GC parameters could be optimized to resolve these peaks and/or broaden peaks to achieve more voltage scan cycles across them. It is theoretically possible to run replicate samples with different voltage scanning programs to achieve a voltage scan with more levels across a narrow peak, but such an approach is of course time-consuming. However, in many applications of field-deployable GCs such as TAG, many chromatograms may be collected in succession that contains similar analytes, potentially allowing one or more complete voltage scans to be recreated. We have mentioned some of those limitations on Line 182 and some further revisions are done to add the limitations:

"We note that voltage switching and data acquisition rates are likely instrument specific and optimal settings may vary significantly across different CIMS instruments. Additionally, some peaks might be too small to finish a complete voltage scan and consequently cannot yield a reasonable $dV_{50}$ unless multiple chromatograms containing the same compounds are collected."

Comment 8: Finally, having used I- CIMS for many years, I am curious about the identities of the non-iodide-adduct ions (i.e. the usually open-shell ion formulas without iodide). This experiment might be an avenue to say something comprehensive about how to interpret these ions. Did you look into these measurements? I understand this is probably outside the scope of this particular manuscript, but would like to know if you have any observations or plans to investigate this topic.

Response: We appreciate the reviewer's insights in taking this work a step further to probe the non-iodide-adduction ions. It is indeed true that the coupling of a GC, the TAG in this study, can be applied to investigate ionization chemistry by the separation of analytes from a mixture. If a chromatographic peak

of a compound is well-resolved in CIMS, all signals detected from the ionization of a single analyte are observed at the same chromatographic retention time and unambiguously assignable to that specific compound, including iodide adducts, products of adduct declustering, fragments (generally not from iodide clustering), and any ions produced by simultaneous alternate chemistry with other ions present in the atmospheric pressure interface (e.g., air). This provides a clean mass spectrum for each chromatographically well-resolved analyte and is particularly useful when analytes are in a complex mixture. In fact, we have obtained some preliminary results and published them in Bi et al., (2021). Here is an excerpt of the work in case the reviewer is interested in the results:

"While the iodide-adduct ions do exist in the mass spectrum of individual analytes, we also observe high abundance of non-adduct ions such as [M-H]- and [M+O2]-. Although such high abundance of [M-H]- may be partially resulted from the tuning-driven declustering of low-polarity adduct ions, the observed non-adduct ions likely account for many ions in the non-adduct region of the iodide valley. By separating analytes chromatographically, these non-adduct ions can be used for the identification of some compounds. These non-iodide ionization pathways can be further enhanced by the intentional introduction of multiple reagent ions.

A multi-reagent ionization mode is investigated in which both zero air and iodide are introduced as reagent ions, to examine the feasibility of extending the use of an individual CIMS for detection of a broader range of analytes. While this approach reduces iodide-adduct ions by a factor of two, $[M-H]^-$ and $[M+O_2]^-$ ions produced from less polar compounds increase by a factor of five to ten, improving their detection by CIMS. The method expands the range of chemical species, which can be measured by CIMS without losing the advantage of identifying chemical formula using the iodide adducts. This novel multi-reagent approach is made possible by combining GC and CIMS detection together with co-measurements from FID. The advantage of simultaneously measuring FID signal for isomer-resolved quantification of I-CIMS sensitivity will be discussed in more detail in a forthcoming paper. Thus, taken together, the GC-CIMS/FID instrument not only inherently valuable for its resolution of isomers in complex atmospheric samples, but also for its ability to characterize and calibrate known CIMS chemistries and to investigate novel and complex chemistries."

Qualitatively, we also observed some non-adduct ions varying in their voltage scanning behavior. We did not systematically investigate such behaviors in this study, but it might indeed be an interesting piece of information to pursue in the future.

Specific/minor comments:
Comment 9: Lines 140-151 (CIMS configuration):
          The pressure, humidity, reagent flow, and voltage configuration (in standard operating condition) of the CIMS should be restated here, since other I- CIMS operators will likely use this paper as a guide for configuring their own instrumentation.

Response: We thank the reviewer for the helpful suggestion. We have added the operating conditions in standard operating mode on Line 143

"Iodide ions are generated by passing a 2 slpm flow of humidified ultrahigh purity (UHP) $N_2$ over a permeation tube filled with methyl iodide and then through a radioactive source (Po-210, 10 mCi, NRD) into the IMR. The IMR pressure is maintained at 100 mbar. Voltages for the ion transfer optics are instrument-dependent due to slight differences in geometry, so we recommend that other users tune the voltages to maximize sensitivity for a weak iodide adduct while minimizing the voltage gradient, which is the tuning approach taken in this study."

Comment 10: Lines 170-185:

The discussion of timing here is confusing. If I understand correctly, the difficultly lies in measuring quickly enough to capture the elution of a single chromatographic peak. What is this timescale and what is the actual measurement speed required? Additionally I don't understand the reasoning behind measuring every 50ms and then averaging three data points. Why not record a measurement every 150 ms?

I think Figure 1 would be improved by showing a time-series of the voltage settings, rather than the blue numbers.

Response: We agree that this section is somewhat confusing and technical and have tried to revise it. The reviewer understands correctly that the difficulty lies in measuring quickly enough to capture the elution of a single chromatographic peak. However, we found the signal in our CIMS takes some time to reach a stabilized level after changing the dV. Therefore, the scan needs to be fast enough to capture the peak yet cannot be too fast to lose stabilized signals under a given dV. One of the reason for the long discussion of timing is that this timescale likely vary significantly across different CIMS and it may depend on instrumental settings and the hardware configurations of iodide-CIMS. Instead of just showing a universal timescale in this study, we would hope to describe the reasoning behind making the decision of measuring every 50 ms and scanning every 400 ms. However, the current discussion might be overly technical to address the need of selecting good measuring frequencies. We have revised the text to improve clarity on Line 174.

"This timescale is not practical for GC applications, in which chromatographic peak widths are typically less than tens of seconds.  In this work, ~~The instrument used in this work was able to switch and stabilize all voltages in ~100 ms. Using these conditions as an example, we develop a rough understanding of the maximum possible rate of voltage scanning. If voltages are varied at 5 Hz (i.e., 200 ms per voltage level), and data is collected at 10 Hz (i.e., 100 ms per data point), data would alternate between single points of "transition spectra" that must be ignored, and single points of real spectra representing the new voltage level. Such an approach is impractical, however, as each level would be represented by a single spectrum, collected immediately after the transition. Instead, to achieve a reasonable and accurate relationship between the signal fraction remaining and the dV, we switch voltages at a slower rate (2.5 Hz) and acquire data at a faster rate (20 Hz~~Generally, the voltages need to be switched fast enough to have multiple dV levels across one chromatographic peak, but slow enough to reach a steady state before switching again. Data acquisition must occur faster than voltage switching, with sufficient time resolution to discard any data collected during voltage transitions (i.e., non-steady-state). In this study, these requirements were met by switching voltages at 2.5 Hz and acquiring data at 20 Hz. With this acquisition rate, eight (i.e. 20 Hz/ 2.5 Hz) data points per dV were collected, with at least the first 2-3 as "transition spectra" that need to be ignored; practically speaking we find that only the last 3-4 spectra are stable (typical relative standard deviation < 20%), so the first 5 spectra of each level are ignored and the signal at a given dV level is taken as the average of the final 3 spectra collected. We note that voltage switching and data acquisition rates are likely instrument specific and optimal settings may vary significantly across different CIMS instruments.

The voltage settings of the CIMS in the regular mode are used as a set of baseline values (designated as dV=2 V). Fourteen different sets of voltage settings, each of which has a constant voltage deviation from the baseline values ( -0.5 V to +12 V). The voltage setting is varied at 2.5 Hz, alternating between the baseline values and a set of voltages representing a different dV level. The voltages upstream of the second quadrupole moved simultaneously with the change of dV to maintain a constant electric field gradient across the quadruples and consequently minimize impacts on ion transmission efficiency (Lopez-Hilfiker et al., 2016). As shown in the upper plot of Figure 1a, the set of dV levels is always in the same

order, but is not monotonic, randomized to avoid the influence of potential memory effects on the results. An example of the output data for a signal chromatographic peak is shown in Figure 1a. Grey dots represent all data collected at 20 Hz, while the larger red dots represent the last three spectra at each ΔV level (ΔV = dV – 2), which are used to calculate signal at that voltage scan with the given ΔV level (blue number, offset to the right)."

As suggested by the review, we have also added a time-series of the voltage settings in Figure 1 to improve clarity of the method.

[Figure]

**Figure 1**. Demonstration of the voltage scan method to the GC-CIMS with a) recreation of chromatographic peak (grey dots represent all data collected at 20 Hz, while the larger red dots represent the last three data points at each dV level, which are used to calculate signal at the given dV level; and b) signal fraction remaining at each voltage difference (dV). Data points in light red, excluded from the sigmoidal fit, are signals from each dV level that do not meet quality control metrics.

The reviewer further asked why not record every 150 ms in this study. In principle, we agree this would work. However, such an approach has a few downsides: (1) it requires an in-depth analysis prior to data collection to determine the optimal timing, which was an open question at the start of this study, (2) it is a bit risky as changes in settings could change the optimal timing and we may not have the ability to adapt in post-processing, and (3) the relative standard deviation between these three points is actually a useful metric to confirm the stability of the voltage level, which may be lost if pre-averaging these points. For these reasons, a faster-than-necessary data acquisition rate is preferred if possible, but as noted by the reviewer it is not strictly necessary.

Comment 11: Lines 193-196:

Wouldn't it make more sense to normalize to the maximum signal, rather than the signal at the arbitrary base voltage = 2? Presumably there are some compounds that do not experience the maximum signal at V=2, but rather at lower voltages.

Response: The reviewer is correct that some compounds may experience maximum signal at voltages lower than 2 V. As shown in Figure 1b, the fitted signal fraction remaining is slightly greater than 1 at dV=2 V. This implies that if we can decrease the voltage difference further, an even higher sensitivity can be observed. Although dV=2V is arbitrary, it is about the minimum voltage difference we can use to ensure basic operations of this instrument. The maximum signal, if exists at an even lower voltage difference, is not necessarily needed to be obtained to calculate $dV_{50}$. More importantly, since one of the objectives in this study is to validate the voltage scan approach for a broader range of chemical species, we follow similar methods used in Lopez-Hilfiker et al. (2016), which firstly proposes this method. In this study, they observed similar results that the normalized signals can exceed 1 at voltage differences lower

than the instrumental settings (e.g., pinonic acid), which indicates that a weaker declustering condition is possible and the sensitivity can be increased even higher by optimizing the instrumental conditions. This theoretical maximum sensitivity can be obtained by multiplying the current sensitivity by the sigmoidal fitted maximum. In summary, while we agree with the reviewer that maximum signal may occur at a even lower voltage settings, we believe that fitting the curve to obtain $dV_{50}$ is independent from the choice of normalized signal.

Comment 12: Lines 268-270:
This approach could possibly bias your results against compounds with particularly low CIMS sensitivity (where the CIMS peak was too small to meet your section criteria) or with particularly high CIMS sensitivity (where the FID peak did not meet the criteria). Were there many peaks that were rejected, and were they mostly CIMS peaks or FID peaks?

Response: We acknowledge that this is one of the limitations of this study. The combination of FID and CIMS means that only compounds can be quantified by both detectors will have comparisons results. Generally, CIMS peaks are more quantifiable because CIMS chromatograms can be separated based on mass to charge ratio of target ions. In contrast, FID is a single channel detector thus making peak resolving more difficult than a CIMS. Therefore, peaks rejected are mostly FID peaks. Of the 82, 142, 158 peaks identified in CIMS in triplicate limonene-O3, limonene-OH, and TMB-OH experiments, 36, 50, and 40 peaks can be quantified by the FID, respectively. We have mentioned this limitation on Line 330 and further discussions on these limitations have been added:

"However, some of those isomers are not included in the discussion because no FID peaks or well-resolved FID peaks are present at the same retention time as their CIMS peaks. Conversely, it is possible that some of the FID peaks are isomers of this formula that are not detectable by CIMS. Due to the higher chemical resolution of the CIMS, the number of isomers available for intercomparison is primarily limited by the chromatographic resolution of the FID since FID is a single channel detector. This limitation can be mitigated by collecting data under a wide range of conditions or environments. Once the CIMS sensitivity of a compound is obtained, quantification can be achieved for those compounds in other poor-signal conditions or even without the coupling of the FID."

Comment 13: Figure 3: It isn't clear whether this figure shows the absolute distribution of sensitivities, or the distribution scaled by the signal abundance of each isomer.

Response: Figure 3 shows the absolute distribution of sensitivities as stated by the y-axis, which is provided in absolute units (ions/mole/million reagent ions). As stated in the caption of Figure 3, the marker size is proportional to the mole fraction of the isomer within a formula. We are not clear on where the ambiguity lies but would be happy to make changes to the figure if the reviewer can provide additional clarification.

Comment 14: Figure 4:
I don't think it is very useful to size the markers by the molar abundance, since there are four distinct experiments shown here, and it is not very meaningful to compare the molar abundance of specific compounds between different experiments (and especially not between the oxidation experiments and the liquid standards).

Response: We agree with the reviewer that across-experiment comparison of molar abundance is not meaningful. However, since the three oxidation experiments are color-coded, we believe it is still

valuable to compare molar abundance within each oxidation experiment. The marker sized by the molar abundance was intended to show that compounds significantly deviated from the log-linear fit still have relatively high abundance and consequently, their presence in the sample mixture cannot be ignored. Simply using the more sensitive isomer to conduct quantitative analysis in direct-air-sampling CIMS may significantly underestimate less sensitive isomers.

Comment 15: Lines 377-380:
Are there any unifying features of the compounds with lower sensitivities? At line 385 you state they are less sensitive compounds, so they may be overwhelmed by the stronger signal from more sensitive compounds. I don't understand this – do they actually have lower sensitivity, or is this just an artifact? And was not the point of the GC pre-separation to be able to consider high-sensitivity and low-sensitivity isomers separately? Additionally, I think it is somewhat misleading to call these "outliers" when they comprise a quarter of the overall dataset.

Response: Unfortunately, we did not identify any unifying features of the compounds that deviate significantly from the log-linear relationship, though the molecular structure and functional groups of those compounds cannot be determined from our data. We believe that future work is needed to probe the chemical properties of those compounds.

The reviewer further asked whether the less sensitive compounds actually have lower sensitivity or it is just an artifact. We believe this confusion comes from a lack of clarity in our discussions and have tried to clarify as below. The section discussed two scenarios: measurements using 1) TAG-CIMS/FID, basically a CIMS with GC pre-separation, and 2) a typical CIMS with direct air sampling. In the context of TAG-CIMS/FID, the low sensitivity of those compounds is actually measured based on the FID calibration of CIMS sensitivity. However, in the case that only a direct-air-sampling CIMS is used (no GC pre-separation), the instrument cannot separate isomer sensitivity. In this case, the more sensitive compounds will likely dominate the signals and the abundance of less sensitive ones may be underestimated. We have revised the manuscript on Line 403:

"Direct-air-sampling CIMS classifies analytes by elemental formula basis while a TAG-CIMS can differentiate isomers and provide quantification down to the isomer-resolution. The analytes significantly deviated from the log-linear relationship are mostly less sensitive compounds. For a CIMS with direct air sampling, the responses of the less sensitive isomers might be overwhelmed by the signals of more sensitive isomers of the same formula.  Such an outcome would potentially strengthen the observed log-linear relationship but would underestimate the observed mass of that compound (and consequently formula)."

We have also replaced "outliers from log-linear" in Figure 4 with "deviations from log-linear"

[Figure]

**Figure 4.** Relationship between sensitivities and $dV_{50}$ of compounds identified in oxidation experiments as well as liquid standards. Each data point is a compound identified with the marker area representing the moles of the compound. The error bars in y-axis are the standard deviation of sensitivity in triplicate (Limonene-$O_3$) or three different OH level measurements (Limonene-OH and TMB-OH). The error bars in x-axis are the standard deviation of $dV_{50}$ in duplicate measurements.

*: unit converted for direct-air-sampling CIMS using 100 mbar in IMR, 2 slpm sample flow rate, and 2 slpm reagent ion flow rate.

Comment 16: Lines 425-434:
Assigning sensitivities randomly distributed around a mean also does not introduce significant bias overall, so I don't think this says much. Could you not achieve the same result simply by calibrating a single compound known to have average sensitivity, and applying that single calibration factor to all compounds?

Response: We apologize for the vague description if it is the case. In fact, we would like to show that the voltage scanning method, despite having large uncertainty for individual compounds, is still valuable in terms of quantification of summed mass. Particularly in the mass conservation analysis where all mass of analytes are investigated to achieve the mass balance in an oxidation process (Isaacman-Vanwertz et al., 2018). We have revised the sentence on Line 451 to improve the clarity:

"Because the log-linear relationship of voltage scanning provides a reasonable central tendency, its application does not introduce significant bias for quantifying the summation of a class of compounds. The predicted total moles of compounds measured agree well with the measured total moles (shown as rectangular markers in Figure 6a, with errors within 30% within an oxidation system and for all oxidation systems combined), although predicted moles of individual compounds have relatively high errors. …. We conclude that, although using voltage scanning introduces high error into the estimation of sensitivity for individual compounds, the approach provides a reasonable estimate of the summed total abundance."

We appreciate the reviewer's suggestion about calibrating the sensitivity of a single compound. However, given the orders of magnitude variance in isomer sensitivity, slight changes in isomer composition, which may occur at different oxidant levels or under varying ambient conditions, might strongly influence the average sensitivity of a formula. Therefore, it might not be feasible to calibrate analytes based on a single compounds known to have average sensitivity, and some average relationship is likely a more accurate approach.

References:
Bi, C., Krechmer, J., Frazier, G., Xu, W., Lambe, A., Claflin, M., Lerner, B., Jayne, J., Worsnop, D., Canagaratna, M. and Isaacman-VanWertz, G.: Coupling a gas chromatograph simultaneously to a flame ionization detector and chemical ionization mass spectrometer for isomer-resolved measurements of particle-phase organic compounds, Atmos. Meas. Tech., 14, 3895–3907, doi:https://doi.org/10.5194/amt-14-3895-2021, 2021.

Claflin, M., Pagonis, D., Finewax, Z., Handschy, A., Day, D., Brown, W., Jayne, J., Worsnop, D., Jimenez, J., Ziemann, P., de Gouw, J. and Lerner, B.: An in situ gas chromatograph with automatic detector switching between ptr- and ei-tof-ms: isomer-resolved measurements of indoor air, Atmos. Meas. Tech., 14, 133–152, 2021.

Heinritzi, M., Simon, M., Steiner, G., Wagner, A. C., Kürten, A., Hansel, A. and Curtius, J.: Characterization of the mass-dependent transmission efficiency of a cims, Atmos. Meas. Tech., 9(4), 1449–1460, doi:10.5194/amt-9-1449-2016, 2016.

Isaacman-Vanwertz, G., Massoli, P., O'Brien, R., Lim, C., Franklin, J. P., Moss, J. A., Hunter, J. F., Nowak, J. B., Canagaratna, M. R., Misztal, P. K., Arata, C., Roscioli, J. R., Herndon, S. T., Onasch, T. B., Lambe, A. T., Jayne, J. T., Su, L., Knopf, D. A., Goldstein, A. H., Worsnop, D. R. and Kroll, J. H.: Chemical evolution of atmospheric organic carbon over multiple generations of oxidation, Nat. Chem., 10(4), 462–468, doi:10.1038/s41557-018-0002-2, 2018.

Isaacman, G., Kreisberg, N. M., Yee, L. D., Worton, D. R. and Arthur, W. H.: On-line derivatization for hourly measurements of gas- and particle-phase semi-volatile oxygenated organic compounds by thermal desorption aerosol gas chromatography ( sv-tag ), , 1–28, doi:10.5194/amt-7-4417-2014, 2014.

Iyer, S., Lopez-Hilfiker, F., Lee, B. H., Thornton, J. A. and Kurtén, T.: Modeling the detection of organic and inorganic compounds using iodide-based chemical ionization, J. Phys. Chem. A, 120(4), 576–587, doi:10.1021/acs.jpca.5b09837, 2016.

Lambe, A. T., Onasch, T. B., Croasdale, D. R., Wright, J. P., Martin, A. T., Franklin, J. P., Massoli, P., Kroll, J. H., Canagaratna, M. R., Brune, W. H., Worsnop, D. R. and Davidovits, P.: Transitions from functionalization to fragmentation reactions of laboratory secondary organic aerosol (soa) generated from the oh oxidation of alkane precursors, Environ. Sci. Technol., 46(10), 5430–5437, doi:10.1021/es300274t, 2012.

Lee, B. H., Lopez-Hilfiker, F. D., Mohr, C., Kurtén, T., Worsnop, D. R. and Thornton, J. A.: An iodide-adduct high-resolution time-of-flight chemical-ionization mass spectrometer: application to atmospheric inorganic and organic compounds, Environ. Sci. Technol., 48(11), 6309–6317, doi:10.1021/es500362a, 2014.

Lopez-Hilfiker, F. D., Iyer, S., Mohr, C., Lee, B. H., D'ambro, E. L., Kurtén, T. and Thornton, J. A.: Constraining the sensitivity of iodide adduct chemical ionization mass spectrometry to multifunctional organic molecules using the collision limit and thermodynamic stability of iodide ion adducts, Atmos. Meas. Tech., 9(4), 1505–1512, doi:10.5194/amt-9-1505-2016, 2016.

Mattila, J. M., Lakey, P. S. J., Shiraiwa, M., Wang, C., Abbatt, J. P. D., Arata, C., Goldstein, A. H., Ampollini, L., Katz, E. F., Decarlo, P. F., Zhou, S., Kahan, T. F., Cardoso-Saldaña, F. J., Ruiz, L. H., Abeleira, A., Boedicker, E. K., Vance, M. E. and Farmer, D. K.: Multiphase chemistry controls inorganic chlorinated and nitrogenated compounds in indoor air during bleach cleaning, Environ. Sci. Technol., 54(3), 1730–1739, doi:10.1021/acs.est.9b05767, 2020.